# Aerosol absorption by in-situ filter-based photometer and ground-based sun-photometer in a Po valley urban atmosphere

Alessandro Bigi[1], Giorgio Veratti[1,2], Elisabeth Andrews[3,4], Martine Collaud Coen[5], Lorenzo Guerrieri[6], Vera Bernardoni[7], Dario Massabò[8], Luca Ferrero[9], Sergio Teggi[1], and Grazia Ghermandi[1]

[1]Department of Engineering 'Enzo Ferrari'. University of Modena and Reggio Emilia, 41125 Modena, Italy
[2]ARPAE, Regional Environmental Agency of Emilia-Romagna, 40122 Bologna, Italy
[3]Cooperative Institute for Research in Environmental Sciences, University of Colorado, Boulder (CO), 80309, USA
[4]NOAA Global Monitoring Laboratory, Boulder (CO), 80305, USA
[5]Federal Office of Meteorology and Climatology, MeteoSwiss, 1530 Payerne, Switzerland
[6]National Institute of Geophysics and Volcanology, ONT, 00143 Rome, Italy
[7]Department of Physics, Università degli Studi di Milano and National Institute of Nuclear Physics INFN-Milan, 20133 Milan, Italy
[8]Department of Physics, Università degli Studi di Genova and National Institute of Nuclear Physics INFN-Genoa, 16146 Genoa, Italy
[9]GEMMA and POLARIS Centre, Department of Earth and Environmental Sciences, University of Milano-Bicocca, 20126 Milan, Italy

**Correspondence:** Alessandro Bigi (alessandro.bigi@unimore.it)

**Abstract.** Light Absorbing Aerosols (LAA) are short-lived climate forcers with a significant impact on Earth's radiative balance. LAA include dust aerosols, Black Carbon (BC) and organic light-absorbing carbonaceous aerosol (collectively termed as Brown Carbon, BrC), which have been also proven to be highly toxic. In this study aerosol absorption at 5 wavelengths (ranging from ultraviolet to infrared) was monitored continuously by filter photometer during two winter seasons in 2020 and 2021 in the city of Modena (south-central Po valley, northern Italy) at two regulatory air quality monitoring sites, along with other pollutants ($PM_{10}$, $PM_{2.5}$, $O_3$, $NO$, $NO_2$, $C_6H_6$) and vehicular traffic rate. Aerosol Optical Depth (AOD) and other column aerosol optical properties were concurrently monitored at four wavelengths by an AERONET sun-photometer at urban background conditions within Modena. In-situ absorption levels were apportioned both to sources (fossil fuel, biomass burning) and to species (BC, BrC), while columnar absorption was apportioned to BC, BrC and mineral dust. The combined analysis of the atmospheric aerosol and gas measurements and of the meteorological conditions (in-situ and by ERA5 reanalysis) identified the location of potential urban sources for BC and BrC, most likely related to traffic and biomass burning. In-situ data show different diurnal/weekly patterns for BrC by biomass burning and BC by traffic, with minor differences between the background and the traffic urban conditions. AERONET version 3 Absorption Aerosol Optical Depth (AAOD) retrievals at 4 wavelengths allowed the estimate of the absorptive direct radiative effect by LAA over the same period under the reasonable assumption that the AOD signal is concentrated within the mixing layer. AERONET retrievals showed a modest correlation of columnar absorption with PBL-scaled in-situ observations, although the correlation improves significantly during a desert dust transport event that affected both in-situ aerosol and columnar absorption, particularly in the blue spectrum range. Low correlation occurred between the contribution of BrC to aerosol absorption for the in-situ and the columnar observations, with

the BrC contribution being generally larger for in-situ observations. Finally, evidence of a highly layered atmosphere during the study period, featuring significant spatial mixing and modest vertical mixing, were shown by ERA5-based atmospheric temperature profiles and by the large correlation of concurrent AERONET AOD retrievals in Modena and in Ispra (on the NW side of the Po valley, ca. 225 km distant from Modena).

## 1 Introduction

Light Absorbing Aerosols (LAA) include: dust aerosols; soot-like, graphitic, elemental carbonaceous light-absorbing particles qualitatively named Black Carbon (BC). A wide range of experimental techniques are available for the experimental measurement of BC, relying on different properties of LAA. In order to harmonize the terminology used reporting the concentration of this species, the scientific community recommends to report BC observations based on light absorption as equivalent BC (eBC, Petzold et al., 2013). eBC aerosol particles have fairly constant refractive index across the ultraviolet – infrared (UV – IR) range (Moosmüller et al., 2009). The eBC concentrations are converted into light-absorbing carbon mass concentration using the mass-specific absorption cross section (MAC). Another type of LAA is Brown Carbon (BrC, Andreae and Gelencsér, 2006; Laskin et al., 2015) which is the fraction of light-absorbing organic aerosol whose optical properties differ from those of BC, because of their enhancement in absorption towards UV wavelengths.

LAA are short-lived climate forcers ($\sim$ 1 week atmospheric residence time (Forster et al., 2021)) and significantly affect the Earth radiative balance (Bond et al., 2013; Wang et al., 2016a). In terms of global impact, BC was shown to have a positive direct radiative effect at the Top-Of-Atmosphere (TOA) in the range of $0.71 - 0.82$ $\mathrm{Wm^{-2}}$ (Chung et al., 2012; Bond et al., 2013; Lin et al., 2014). Estimates of global direct effect were lower for BrC than for BC, in the range of $0.04 - 0.57$ $\mathrm{Wm^{-2}}$ (Feng et al., 2013; Lin et al., 2014; Saleh et al., 2014; Jo et al., 2016; Brown et al., 2018; Zhang et al., 2020). BrC concentrations are very spatially variable and concentrations depend on the study specifics. Due to aerosol-cloud interactions, the overall effective radiative forcing of LAA (i.e. the difference in their radiative effect between the present day and pre-industrial times (Heald et al., 2014)) ranges between $0.15 \pm 0.17$ $\mathrm{Wm^{-2}}$ for BC (Thornhill et al., 2021; Forster et al., 2021), with the largest part of this uncertainty arising mainly from the indirect and semi-direct effect exerted by aerosol on cloud condensation nuclei, ice nuclei and on the atmospheric lapse rate, along with the aerosol mixing state (Twomey, 1974; Charlson et al., 1992; Bond et al., 2013; Rosenfeld et al., 2014; Takemura and Suzuki, 2019).

In addition to the effects on climate, the scientific literature has documented the adverse effects on human health of aerosol, which significantly affects life expectancy (Loomis et al., 2013; Cohen et al., 2017; West et al., 2016). The toxicological effect of particulate matter (PM) is known to depend on the aerosol size distribution and chemical composition (Pöschl, 2005). BC is one of the components with a proven harmful effect on human health (Janssen et al., 2012), and both long-term and acute exposure to increased eBC concentrations have been shown to increase the mortality risk (Ostro et al., 2015; Yang et al., 2021). Recent studies have also shown that the exposure to increased eBC concentrations were positively associated with various health issues such as ischemic heart disease and myocardial infarction (Luben et al., 2017; Magalhaes et al., 2018; Kirrane et al., 2019). In addition, Regencia et al. (2021) observed that short-term cumulative exposure to traffic-related eBC

concentrations could adversely affect blood pressure, resulting in cardiovascular diseases. BrC has also been shown to have detrimental health effects, enhanced because of its enrichment in organic compounds (Chowdhury et al., 2019; Offer et al., 2022), possibly related to aerosol aging (Li et al., 2022; Tuet et al., 2017; Weitekamp et al., 2020).

The compilation of reliable and accurate emission inventories for eBC is critical for the development of robust air quality control strategies and the mitigation of global warming. However, the large uncertainty associated with source emission factors, PM speciation and eBC definition makes the implementation of systematic and harmonized emission estimates a challenging task. Despite these limitations, most studies identify road transport as the largest eBC emission source in Europe (Wang, 2015), followed by biomass burning and industry (European Environment Agency, 2013), as more recently confirmed by the analysis

of the eBC emission change in Europe due to COVID-19 lockdowns (Evangeliou et al., 2021). Similar to BC, BrC can be directly emitted into the atmosphere during the combustion of fossil fuels, although its major source is biomass burning. BrC can also originate from secondary reactions, e.g. through aging processes or by photo-oxidation of biogenic or anthropogenic volatile organic compounds (Laskin et al., 2015).

Several approaches have been proposed in the literature to measure LAA, including photothermal interferometry, photo-

acoustic spectroscopy, and on-line or off-line filter-based light attenuation methods (Lack et al., 2014). The difference in the BC reported by these techniques increases when significant amounts of secondary organic are present (Kalbermatter et al., 2022). Both the interferometric and the acoustic approaches can be considered thermal based measurements, since they quantify the fraction of absorbed optical energy that is rapidly transferred into the surroundings under a controlled light source emission. The main advantage of these techniques is their direct measurement of the absorption of particles while suspended in air,

however they both suffer from technical and operational limitations. For example, the photo-acoustic technique is very sensitive to atmospheric conditions such as relative humidity, temperature and pressure (Langridge et al., 2013), while photothermal interferometry is sensitive to mechanical vibration, although it has recently gained new attention (e.g. Visser et al., 2020; Drinovec et al., 2022). Filter-based measurements are very simple to operate, but have the main disadvantages of filter-related artifacts, such as the filter loading and the multiple scattering effects within filter fibres and between the collected particles and

the filter fibres, possibly leading to systematic errors in the measurements. With the aim to overcome these limits, different technical and analytical corrections have been developed to correct for the non-idealities of filter-based measurements (e.g. Weingartner et al., 2003; Petzold et al., 2005; Virkkula et al., 2007; Collaud Coen et al., 2010; Hyvärinen et al., 2013; Drinovec et al., 2015; Li et al., 2020), for filter absorption photometers common in field experiments and in air quality monitoring networks. The aethalometer (Magee Scientific Co., Berkeley, USA), is a commonly used filter-based photometer designed to

measure LAA at multiple wavelengths and at high temporal resolution, generally at fixed monitoring sites. Lightweight portable micro-aethalometers, such as the AE51 or the MA200 series (Aethlabs, San Francisco, USA), were recently developed and successfully used in complex urban environments for pedestrian exposure assessments (Viana et al., 2015; Good et al., 2017; Boniardi et al., 2021), mobile observations (Grivas et al., 2019; Liu et al., 2019, 2021) and vertical profile investigations through unmanned aerial vehicles (UAVs) and balloons (Ferrero et al., 2011, 2014; Pikridas et al., 2019; Kezoudi et al.,

2021). Despite their limitations, multi-wavelength aerosol absorption observations by filter photometers have proven suitable for the application of source and component apportionment models, such as the 'Aethalometer model' (Sandradewi et al.,

2008) to apportion BC between wood burning and fossil fuel combustion emissions or the Multi-Wavelength Absorption Analyzer (MWAA, Massabò et al., 2015; Bernardoni et al., 2017) algorithm, which enables disentanglement of the BC and BrC components of LAA, and a determination of their radiative forcing (Ferrero et al., 2021a).

Surface in-situ aerosol measurements can provide important information about aerosol characterization and concentration for the lowest tropospheric layer. However, estimating the vertical distribution of aerosol particles or their columnar load remains crucial to completely understanding their impact on the climate system. In order to meet this need, the worldwide network of calibrated sun/sky photometers AErosol RObotic NETwork (AERONET, Holben et al., 1998) was developed, with the goal of measuring aerosol optical columnar properties, e.g. aerosol optical depth (AOD) and column single-scattering albedo (SSA).

Numerous studies have attempted to compare in-situ observations with ground-based columnar aerosol optical properties providing different results depending on the atmospheric mixing state, the aerosol vertical profile and the local/regional pollution conditions. Several authors used the ratio between the surface in-situ aerosol mass concentration or aerosol absorption and the boundary-layer-height (i.e. they rescaled surface data over this atmospheric layer), and showed how this ratio underestimated sun-photometry observations of AOD or absorption AOD (AAOD) respectively (e.g. Bergin et al., 2000; Slater and Dibb, 2004;

Aryal et al., 2014; Chauvigné et al., 2016; Chen et al., 2019). These findings were consistent across various types of locations (e.g. rural background, moderately polluted or marine) and highlighted that, in those settings, generally the main factors limiting the representativity of surface in-situ measurements of the atmospheric column are the aerosol mixing within the boundary layer (BL) and the presence of aerosol above the BL, which can contribute significantly to the extinction and absorption in the column.

Datasets allowing a worldwide trend analysis in LAA levels remain limited (Laj et al., 2020), however according to both in-situ (Collaud Coen et al., 2020) and ground-based columnar (Li et al., 2014) observations, in the northern hemisphere, particularly in the US and Europe, the aerosol absorption coefficient ($\sigma_{ap}$) decreased over the last decade(s). More specific to the region of interest for our study, the Po valley is a European hot-spot for atmospheric pollution situated in northern Italy. A previous work on the Po basin observed a decrease for both columnar AOD and in-situ aerosol scattering and absorption

in Ispra, on the NW side of the Po valley, in the early 2000s (Putaud et al., 2014). This drop was consistent with a significant valley-wide decrease in $PM_{10}$ and $PM_{2.5}$ in-situ ground levels (Bigi and Ghermandi, 2014, 2016), thanks also to a drop in primary PM emissions by vehicular transport. Similarly, a drop of ~4% per year over the period 1997 – 2016 was recorded for the elemental carbon content in fog samples at the rural background site of San Pietro Capofiume (Gilardoni et al., 2020b).

    Significant aerosol sources other than traffic remain present in the valley, e.g. biomass burning by domestic heating for

several compounds including organic aerosols and BC, and farming for $NH_3$, a major PM precursor. Their role in PM levels was highlighted by the small decrease in PM across the basin (Ciarelli et al., 2021; Putaud et al., 2021) and in particle count in Modena (Shen et al., 2021) during the 2020 lockdown due to the SARS-CoV-2 pandemics. Some studies in the Po valley addressed temporal and vertical variability of $\sigma_{ap}$ in Milan, the largest city of the basin (Ferrero et al., 2011, 2014; Vecchi et al., 2018). These authors found a decline in BC levels within the mixing layer, with higher BC levels observed at the ground (i.e.

50 – 100 m) and a marked drop (more than 50%) above the mixing height, with BC contributing to $\sim$10% ($\sim$8%) of the overall $PM_1$ extinction (mass) at a surface Milan urban background site in winter. Other studies in the Po valley focused on the effect

of the reduction in $NO_x$ and $NH_3$ on $PM_{2.5}$ levels (Veratti et al., 2023), as well as on the impact of biomass burning on surface aerosols, particularly at the rural background site of San Pietro Capofiume and the urban site of Bologna (Gilardoni et al., 2016; Costabile et al., 2017; Paglione et al., 2020). These latter studies highlighted the large Absorption Ångström Exponents (AAE, Moosmüller et al., 2009) for biomass burning organic aerosol, ranging from $\sim 3 - 5$, mainly due to aged aerosols in the aqueous phase and related to an increase in the organic aerosol/BC mass ratio. Previous investigations of the spatial variability of PM surface observations highlighted the impact of large urban areas on aerosol load, particularly for $PM_{10}$, using cluster analysis (Bigi and Ghermandi, 2014, 2016). A Europe-wide assessment of urban air quality by Thunis et al. (2017), based on a simplified dispersion model, estimated a 57% contribution by in-city emissions to urban $PM_{2.5}$ in Milan, making this city the one with the largest self-contribution to local $PM_{2.5}$ across the European Union. Similarly the spatial variability in columnar aerosol load observed by ground-based remote sensing instruments between Ispra and the Adriatic sea, east of the Po basin, showed larger AOD and lower SSA at the Ispra site (Clerici and Mélin, 2008), confirming the impact of in-valley combustion emissions.

Relying on these previous findings, the current study provides additional knowledge on LAA in the Po valley by investigating the temporal, spatial and columnar variability of $\sigma_{ap}$ in Modena, an urban area representative of several cities in the basin. The city of Modena is located in the central-south part of the Po Valley. The study period is winter $2020 - 2021$ and the experimental dataset includes both in-situ and ground-based columnar observations. Additionally, source apportionment of $\sigma_{ap}$ using the in-situ and the ground-based columnar observations in Modena are compared to investigate the impact of low level emissions and long range transport on the aerosol optical properties, together with the first estimation of LAA heating rate (HR) and its diurnal trend in Modena. Finally, more insight into the spatial and temporal variability of the different absorbing components in the Po valley are provided by a comparison between columnar optical properties in Modena and Ispra. Below we first describe the measurements we will use and then address these topics.

## 2  Measurement site and methods

Modena (44.6° N, 10.9° E, 32 m a.s.l., ~ 180 000 inhabitants) is located on the centre-south side of the Po valley, in northern Italy, a basin surrounded by the Alps and the Apennines. The basin area is affected by recurrent atmospheric temperature inversions in winter and low wind conditions, leading to a build-up of atmospheric pollutants. The result is that the Po valley is one of the largest European regions exceeding the daily $PM_{10}$ limits set by the European regulation (EC 50/2008) and by the World Health Organisation (WHO) guidelines (WHO, 2021). The city is situated in flat topography, 13 km north of the foot of the closest Apennine hills and 96 km south of the foot of the Alps, i.e. it is on the Southern side of a wide (~ 110 km) valley.

The latest bottom-up regional emission inventory for the area of the municipality of Modena (ARPAE, 2020), reference year 2017, identifies traffic and domestic heating as the main $PM_{10}$ sources, contributing 38% and 58% of total emissions respectively, although Modena also hosts a few districts for light manufacturing (Selected Nomenclature for sources of Air Pollution, SNAP 3 and 4), contributing 3% of total $PM_{10}$ emissions (Figure 1). More specific to non-industrial combustion

(SNAP 2), most of buildings use compressed natural gas for both heating and cooking; consistently 99.4% of $PM_{10}$ emissions
by SNAP 2 are estimated to be produced by biomass combustion for domestic heating (ARPAE, 2020).

As is common to most urban areas in the basin, vehicular traffic is the main source of $NO_x$ emissions (78% of total $NO_x$ emissions, ARPAE, 2020), with a significant impact on local air quality (Ghermandi et al., 2020; Veratti et al., 2020) and on population exposure (Veratti et al., 2021). Modena's setting is quite representative of several mid-size urban areas across the Po valley, particularly in terms of traffic and domestic emissions sources and topography.

## 2.1 In-situ surface measurements

Two MA200 micro-aethalometers were installed in Modena, sampling from the gently heated (~ 30 ±2 °C) glassware manifold inlet lines already in use for reactive gas monitors at the two regulatory air quality monitoring sites in town: Giardini (EoI code IT0721A, 44.637° N, 10.906° E, 39 m a.s.l.) and Parco Ferrari (EoI code IT1771A, 44.652° N, 10.907° E, 30 m a.s.l). These two sites are representative of urban traffic and urban background conditions, hereafter referred as UT and UB respectively (see Figure 1 for their location). The UT site faces a major road with two lanes per direction, with estimated median daily traffic counts of ~20 thousand vehicles, while the UB is within Modena's largest urban park at a distance of ~120 metres from the nearest road. The inlet height at both sites is approximately 4 m above ground. The inlet has no size cut, i.e. the instruments are sampling total suspended particles.

The MA200 are filter absorption photometers measuring at 5 wavelengths ($\lambda$ = 375 nm, 470 nm, 528 nm, 625 nm, 880 nm) using PTFE filter tapes. AAE for in-situ observations (hereafter AAE[i]) was computed by a fit to absorption at all 5 wavelengths. The instruments were used in dual-spot sampling mode (firmware 1.09 and 1.10 were installed during the study) and a compensation algorithm similar to the one proposed by Drinovec et al. (2015) is applied by the internal firmware. This firmware uses a multiple scattering correction coefficient $C_{ref}$ = 1.3, which was chosen by the manufacturer in order to mimic the response by the Aethalometer AE33 (Aethlabs, personal communication).

Aerosol absorption monitoring at the UT site was performed between 19 January – 21 April 2020 and 9 November 2020 – 8 March 2021 with a time resolution of 1 minute. At the UB site, aerosol absorption was monitored between 4 February 2020 – 13 October 2020 at 1 minute time resolution and between 13 December 2020 – 20 March 2021 at a 5 minute time resolution. In order to compensate for the occasionally low absorption readings at the latter site, the 1-minute raw transmittance counts at UB were firstly aggregated to 5 minutes and then used to compute the corresponding $\sigma_{ap}$ by a transcription in R programming language of the dual-spot compensation algorithm as described in Drinovec et al. (2015). The MA200 measurements were screened depending on the status reported by the instrument. Flow calibration was performed before each filter change. Flow was set to 100 $\mathrm{ml\,min^{-1}}$ in winter and increased to 125 $\mathrm{ml\,min^{-1}}$ in summer, because of the lower atmospheric concentrations. In the present study only measurements from winter months were analyzed, i.e. December, January and February. Strict lockdown restrictions in Northern Italy due to the SARS-CoV-2 pandemic lasted from 8 March 2020 until 4 May 2020, therefore the winter data reported here are representative of a business-as-usual scenario, partly spanning across two winter seasons. Absorption data were averaged to 1 hour prior to the analysis, in order to match the time resolution of other analyzed variables.

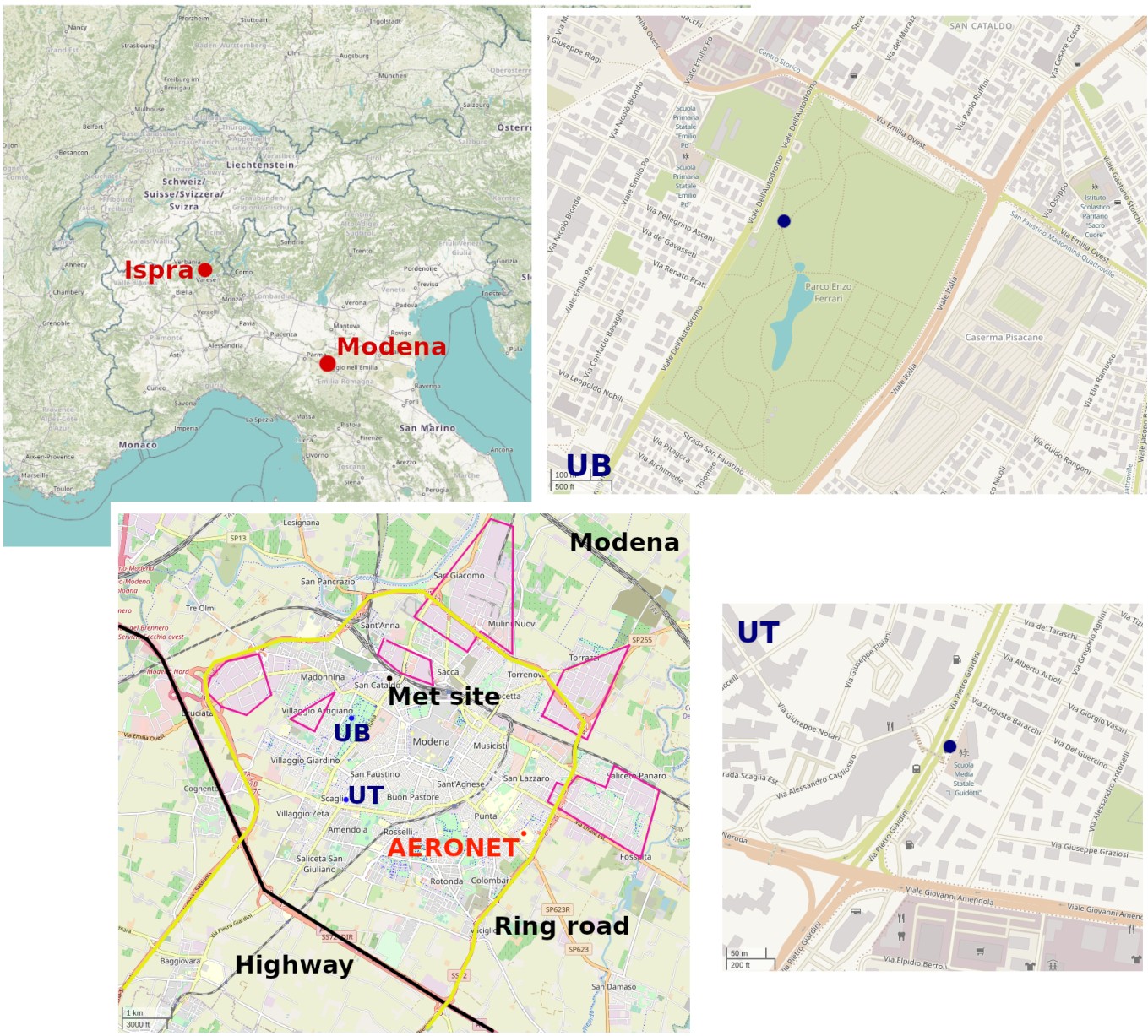

**Figure 1.** Setting of the measurement site. Areas outlined in purple indicate the manufacturing districts (from © OpenStreetMap contributors 2023. Distributed under the Open Data Commons Open Database License (ODbL) v1.0.).

A comprehensive uncertainty analysis for absorption observations by the MA200 has not yet been fully performed by the scientific community. Li et al. (2021) looked at the multiple scattering uncertainty of its PTFE filter and suggested that multiple scattering artifacts might lead to an overestimation of the absorption by BrC, with the bias dependent on the absorbing

190

strength of the compound, i.e. a behaviour qualitatively similar to that of the AE33 (Yus-Díez et al., 2021). Alas et al. (2020) performed a large intercomparison involving these devices, testing several MA200s (single spot, 10 s and 60 s time resolution) and highlighted a low unit-to-unit variability (ca. 2%) across all wavelengths and good agreement ($R^2 > 0.93$) for loading-corrected eBC when compared to the AE33. In the current study a 8% uncertainty was attributed to MA200 absorption, based on the mean standard error of the slope of the linear regression between the eBC by the MA200 and the AE33 found by Alas et al. (2020).

Regulatory air quality data were also available at the two sites and include NO, $NO_2$, $O_3$ (UB site only), and $C_6H_6$ (UT site only) at hourly time scale. $PM_{10}$ and $PM_{2.5}$ (at UB site only) were also available, although on a daily time scale. The daily $PM_{10}$ median (10th, 90th quantiles) concentration at the UB site over the period 2017 – 2021 was 24 $\mu g\,m^{-3}$ (13 $\mu g\,m^{-3}$, 57 $\mu g\,m^{-3}$), while at the UT site the same statistics for $PM_{10}$ were 27 $\mu g\,m^{-3}$ (14 $\mu g\,m^{-3}$, 63 $\mu g\,m^{-3}$). Consistently, over the same period, hourly $NO_2$ at the UB showed lower levels than at the UT site, with the two locations having a median (10th, 90th quantiles) of 23 $\mu g\,m^{-3}$ (6 $\mu g\,m^{-3}$, 50 $\mu g\,m^{-3}$) and 35 $\mu g\,m^{-3}$ (14 $\mu g\,m^{-3}$, 66 $\mu g\,m^{-3}$) respectively.

Direct traffic counts were also available for the urban area during the period of investigation. These data were collected by 400 induction loops for traffic light control within the urban and suburban street network. Continuous vehicle counts from the induction loops nearest to the UT and UB sites were aggregated into one hour total traffic data; these hourly aggregates were used primarily to highlight variability in traffic patterns. The uncertainty in the count by these devices is approximately 10% (Bellucci and Cipriani, 2010).

Meteorological variables were provided by the regional weather monitoring network station within the urban area of Modena, and include wind speed and direction (WS, WD), atmospheric temperature (*T*), relative humidity (RH), downward global radiation (*Q*) and atmospheric pressure (*p*). The site is on the roof of the municipality offices at 40 meters above ground and is the highest weather station in the urban area with data available over the study period. These data provide indications of the wind conditions inside the urban canopy, but may differ from wind conditions at the 4 m height of the MA200 measurement.

## 2.2   Mixing layer height

Hourly estimates for the mixing layer height (MLH) were provided by ERA5 reanalysis (Hersbach et al., 2018). ERA5 reanalysis, provided by the European Centre for Medium-Range Weather Forecasts (ECMWF), proceeds from the data assimilation of global observations into the Integrated Forecast System (IFS), a global numerical weather prediction model, to produce a globally complete and consistent dataset of physical quantities, continuous in time and space. ERA5 provides hourly estimates for several geophysical quantities, including MLH, at a grid resolution 0.25° × 0.25° over the period 1940 - today. For this study we extracted the MLH at the ERA5 grid point with coordinates 11.00° W, 44.75° N. Modena is about 14 km south of this grid point, i.e. it lays between two ERA5 grid points, since in this region the size of the ERA5 cell is ~20 km × ~28 km: MLH was extracted at that grid point since it is representative of a cell over an area with flat topography, i.e. very similar to the area of Modena. MLH estimates by ERA5 are used in the analysis, since no experimental estimates of the planetary boundary layer height were available in town and the closest location with regular atmospheric sounding (at 12 UTC and 00 UTC) is in San Pietro Capofiume, a rural background site surrounded by a flat topography 53 km east of Modena. The ERA5 estimate of

MLH in Europe was assessed to be underestimated on average (median) by ~54 m (~19 m) based on a comparison between ERA5 and daytime radiosoundings by Guo et al. (2021). This underestimate represents a lower end estimate, since generally soundings in Europe are taken around 12 UTC, i.e. when the MLH is quite developed.

It is worth noting that deficiencies have been observed in various planetary boundary layer and surface parameterizations for conventional meteorological models (e.g. IFS, WRF, and COSMO. Martilli et al., 2021; Maroneze et al., 2021; Lapo et al., 2019; Battisti et al., 2017), leading to a challenging characterization of strong thermal inversions (Mahrt, 2014; Acevedo et al., 2019), like those occurring in the Po valley. All these meteorological models typically rely on a single stability parameter, e.g. the Richardson number or the turbulent kinetic energy, to automatically estimate MLH based on fixed thresholds, regardless of the wide range of possible atmospheric conditions, since these models cannot apply *ad-hoc* methods for each individual situation. More specific to this study, MLH estimates by IFS are based on the bulk Richardson number (Vogelezang and Holtslag, 1996), regardless of the atmospheric stability conditions, and MLH is defined as the lowest level at which the bulk Richardson number reaches the critical value of 0.25 (ECMWF, 2017).

## 2.3 Ground-based columnar measurements

Column-integrated measurements of optical properties of the Modena urban atmosphere were collected by a multi-channel Cimel CE-318 sun/sky photometer installed on the roof of the Dept. of Engineering 'Enzo Ferrari' at about 20 m above the ground. The instrument is part of NASA's AERONET network (Holben et al., 1998). This site, within the grounds of the University campus and representative of residential background conditions, is on the southeastern edge of the urban settlement, while the UB and UT sites are on the west side of the town at a distance of 4 km and 3.5 km respectively (Figure 1).

In our analysis of the Cimel data, we considered both Level 2.0 and Level 1.5 version 3 almucantar retrievals at 4 wavelengths ($\lambda$ = 440, 675, 870, and 1020 nm) (Sinyuk et al., 2020). Level 2.0 absorption data are more robust (e.g., Dubovik et al., 2000), but Level 1.5 data provide more matches with surface measurements, as discussed below. The almucantar retrievals provide several columnar properties including Absorption Aerosol Optical Depth (AAOD), SSA, the depolarization ratio and the lidar ratio at the 4 wavelengths, as well as the particle volume size distribution and AOD apportioned to submicron and supermicron aerosols from which fine mode fraction (FMF) is calculated (O'Neill et al., 2003). AERONET retrievals also allowed estimation of the Scattering AOD (SAOD($\lambda_j$) = Total AOD($\lambda_j$) - AAOD($\lambda_j$)) for each of the 4 wavelengths, and the wavelength dependence of SAOD, i.e., the column Scattering Ångström Exponent (hereafter SAE[c]), as well as the column Absorption Ångström Exponent (AAE[c]).

There is much information in the literature about the uncertainty in AERONET products (e.g. Eck et al., 1999; Andrews et al., 2017; Sinyuk et al., 2020; Kayetha et al., 2022). A fixed AOD uncertainty set to 0.01 was used, following Eck et al. (1999). For the current study, AOD-dependent uncertainty in SSA at 440, 675 and 875 nm by AERONET v3 retrievals was estimated based on the data from a urban site in Sinyuk et al. (2020), ranging from 0.017 at 440 nm when $AOD_{440}$ is 0.7 to 0.103 at 870 nm when $AOD_{440}$ is 0.03. The overall uncertainty for the analysed aerosol parameter, e.g. AAOD, was estimated as a propagation of the uncertainties of AOD and SSA and ranged between 0.011 and 0.033.

In addition, the direct radiative effect (DRE) at the top of the atmosphere (TOA) and bottom (BOA), retrieved by AERONET in clear sky conditions in winter, were considered. Since the atmospheric aerosol is characterized by a significant absorptive capacity the difference between the DRE at TOA and BOA (hereafter $\Delta$ DRE$_{atm}$) represents the instantaneous radiative power density absorbed along the atmospheric column by the aerosol within that atmospheric layer (Chakrabarty et al., 2012; Kedia et al., 2010). $\Delta$ DRE$_{atm}$ is expressed in $\mathrm{W\,m^{-2}}$, which is the common metric used in the literature to quantify the integrated radiative power density absorbed by the aerosol in the atmosphere (Kedia et al., 2010; Das and Jayaraman, 2011; Bond et al., 2013; Heald et al., 2014). However, as demonstrated in Ferrero et al. (2014), a more useful parameter is the Absorptive DRE (ADRE) of atmospheric aerosol, which can be computed simply by normalizing $\Delta$ DRE$_{atm}$ by the atmospheric thickness $\Delta z$ hosting most of the LAA, and this thickness in the Po valley can be generally assumed to correspond to the MLH. The ADRE represents the radiative power absorbed by the aerosol for unit volume of the atmosphere ($\mathrm{W\,m^{-3}}$). The advantage of using ADRE in the Po Valley environment in wintertime is that, in this case, most of the AOD signal is built up within the mixing layer, as shown by both Ferrero et al. (2019), who found that in Milan up to 87% of AOD signal was generated within mixing layer, 8% in the residual layer and 5% in the free troposphere, and by Barnaba et al. (2010), who found similar figures at the Ispra background site. This means that if the thickness $\Delta z$ is the MLH, the ADRE will refer to that layer with an expected maximum overestimation of approximately 13% (i.e. roughly the amount of aerosol optical depth above the MLH). From the ADRE the instantaneous heating rate (HR, $\mathrm{K\,day^{-1}}$) can be computed as (Ferrero et al., 2014):

$$HR = \frac{\mathrm{ADRE}}{\rho\,\mathrm{C_p}} \tag{1}$$

where $\rho$ is the air density and $\mathrm{C_p}$ ($1005\ \mathrm{J\,kg^{-1}\,K^{-1}}$) is the isobaric specific heat of dry air. The most important advantages of this AERONET-based approach to derive the LAA HR are: (a) the possibility of obtaining a rapid HR estimation to investigate the HR temporal evolution during a selected time period and (b) the possibility of deriving the HR using a well-established network (AERONET) allowing a global comparison of the output. This approach is limited because HR can be obtained directly by the AERONET retrievals only if most of the AOD signal is built up within the mixing layer (thus with an expected overestimation of $\sim$13%). Due to these limitations, the analysis of the HR is limited to retrievals collected during days without significant dust content. This screening process followed the same process used for in-situ data described in section 2.4.1, since the apportionment of in-situ data suffered from a similar limitation as that of the HR analysis.

Furthermore, in some of the analysis described below, the in-situ and columnar data were compared, requiring temporal matching of the two data sets. An in-situ/columnar observation match is considered successful when the AERONET retrieval occurred during the hourly averaged in-situ measurement. Level 1.5 version 2 AERONET data are known to have large uncertainty when AOD at 440 nm (AOD$_{440}$) is less than 0.4 (Dubovik et al., 2000). In order to maximize the availability of columnar measurements for the analysis, Level 1.5 data were used. Level 1.5 data points with AOD$_{440} \leq 0.2$, were discarded from the analysis and the data remaining after the AOD screening are referred to as L1.5* in what follows.

## 2.4 Source apportionment of in-situ and columnar data

Both in-situ and columnar data were apportioned according to the aerosol spectral properties, i.e solving the balance of the aerosol absorption based on its dependence on the AAE, on the absorbing species and on the wavelengths. Two different apportionment approaches were used for the in-situ and the columnar observations, although based on the same foundation. The approach applied to the in-situ data requires at least five wavelengths to ensure stability (Bernardoni et al., 2017).

### 2.4.1 In-situ apportionment

In-situ aerosol absorption $\sigma_{ap}$ was apportioned to species (Black Carbon, Brown Carbon, referred to as $\sigma_{ap}^{BC}(\lambda)$ and $\sigma_{ap}^{BrC}(\lambda)$ respectively) and sources (fossil fuel and biomass burning combustion, referred to as $\sigma_{ap}^{ff}(\lambda)$ and $\sigma_{ap}^{BB}(\lambda)$) using the Multi-Wavelength Absorption Analyzer model (MWAA, Massabò et al., 2015; Bernardoni et al., 2017). This model assumes an equivalence between the Absorption Ångström Exponent (AAE, Moosmüller et al., 2009) of BC and that of fossil fuel ($AAE^i_{BC} = AAE^i_{ff}$), and it assumes biomass burning to be the only source of BrC. Under these hypotheses, the MWAA model assumes

that both the following equations hold for the total $\sigma_{ap}(\lambda)$ at each wavelength:

$$\sigma_{ap} = \sigma_{ap}^{BC}(\lambda) + \sigma_{ap}^{BrC}(\lambda) = A \cdot \lambda^{-AAE^i_{BC}} + B \cdot \lambda^{-AAE^i_{BrC}} \tag{2}$$

$$\sigma_{ap} = \sigma_{ap}^{ff}(\lambda) + \sigma_{ap}^{bb}(\lambda) = A' \cdot \lambda^{-AAE^i_{ff}} + B' \cdot \lambda^{-AAE^i_{bb}} \tag{3}$$

  In Equations 2 and 3 $AAE^i_{BC} = AAE^i_{ff} = 1$ was set, based on the $AAE^i$ computed over 5 wavelengths at morning rush hour on winter weekdays at UT, consistent with fresh uncoated BC particles (e.g. Liu et al., 2018). $AAE^i$ for BrC was determined

by a preliminary non-linear fit of Equation 2, performed considering $AAE^i_{BrC}$ as a free parameter (and resulting in an average $AAE^i_{BrC} = 3.9$); $AAE^i_{bb} = 2$ was set based on literature data for the Po valley (Bernardoni et al., 2011, 2013; Costabile et al., 2017; Vecchi et al., 2018). A and B were then obtained for each sample by multi-wavelength fit of Equations 2 (after fixing $AAE^i_{BrC}$) and A', B' by multi-wavelength fit of Equations 3. It is noteworthy that the MWAA model neglects possible contributions from mineral dust. To limit uncertainties resulting from this, the days with significant dust load were discarded prior

the application of the MWAA model to the in-situ data, i.e. whenever the in-situ apportionment data is presented throughout the text, it is screened for dust. Days with significant dust content were first identified for the atmospheric column, using the particle volume size distribution estimated by the AERONET inversion (Sinyuk et al., 2020); the identification of dust events was performed qualitatively, based on the retrievals having a dominant coarse mode (e.g. Figure S7, panel b). These retrievals were subsequently double-checked by 72-hours HYSPLIT back trajectories using Global Data Assimilation System (GDAS)

1° resolution wind fields. Additionally, the impact of dust at ground level was assessed based on the daily $PM_{2.5}$ to $PM_{10}$ ratio from the in-situ measurements (Figure S1), with ratio $\leq 0.5$ as a qualitative threshold for a dust event. For reference, the daily $PM_{2.5}$ to $PM_{10}$ ratio in winter between 2017 to 2021 at the UB site had a median ratio of 0.71 and a 10th (25th) quantile of

0.53 (0.62), i.e. the two aerosol fractions are quite similar, as previously observed at most UB sites across the basin (Bigi and Ghermandi, 2016).

### 2.4.2 Columnar apportionment

AAOD was apportioned to BC, BrC and mineral dust using the approach proposed in Bahadur et al. (2012), i.e. by directly solving the system of Ångström equations (see Appendix A) using the AERONET almucantar L1.5* retrievals. The system includes Equation A1, describing the additive contribution of AAOD by each species to the total AAOD and Equation A2, describing the exponential dependence of AAOD on the wavelength. This apportionment method neglects the mixing state of absorbing species (i.e., the aerosol is assumed to be externally mixed), and assumes the observed AAOD is representative of a well-mixed sample of these species. Bahadur et al. (2012) estimated globally valid ranges of $AAE^c$ and $SAE^c$ for BC, BrC and dust, parameters needed to solve the system of AAE equations, based on long-term, worldwide AERONET observations (version 2, level 2.0).

For the current study, a tailored estimate of the $AAE^c$ values for Modena was performed, based on the full time series of AERONET retrievals in Modena (from Jan 2000 to June 2021). The classification of aerosol species (BC, BrC, dust) in order to estimate their $AAE^c$ values was performed by combining the approaches by Bahadur et al. (2012), Cazorla et al. (2013) and Shin et al. (2019). Cazorla et al. (2013) suggests threshold values in $SAE^c$ and $AAE^c$ across the 440 – 675 nm range (hereafter $SAE1^c$ and $AAE1^c$), which were applied for a preliminary classification (Figure S2). Shin et al. (2019) combined the particle linear depolarization ratio and the lidar ratio at 1020 nm into a dust ratio coefficient $\chi_{d,\lambda}$, estimating the contribution by dust and non-dust aerosol to AOD. Following Bahadur et al. (2012), in order to disentangle the spectral properties of fossil fuel and biomass burning aerosol, first $AAE^c$ for dust was assessed using the full L1.5* time series (259 data points), based on the conditions $SAE1^c < 1$, $AAE1^c > 1.5$ and $\chi_{d,1020nm} > 0.8$. Since the major source of biomass burning in the Po valley is domestic heating during winter. $AAE^c$ for BC was estimated based on the full time series of summer L1.5* retrievals (1752 data points). The conditions applied in this case were $SAE1^c > 1.2$ and $AAE2^c/AAE1^c > 0.8$, with index 1 indicating th range 440 – 675 nm and index 2 indicating the range 675–880 nm (Bahadur et al., 2012). Then the $AAE^c$ for BrC was computed by solving the AAE equations system on the L1.5* non-dust winter retrievals over the period 2015-2022 (89 data points). $AAE^c$ for BC and BrC are based on datasets with different size since in winter fewer retrievals are available, due to shorter daytime duration and clouds: to limit the possible bias induced by this difference in sample size and by the potential presence of outliers, the median ± median absolute deviance of the $AAE^c$ for dust, BC and BrC were computed, for both Modena and Ispra (see below), and reported in Table 1. Table 1 also includes literature values of column $AAE^c$ for different absorbing aerosol types for comparison.

To assess the representativity of the $AAE^c$ values derived for Modena, sun/sky photometer retrievals in Modena were also compared to AERONET data from Ispra (45.80° North, 8.63° East, 220 m a.s.l., 225 km NW of Modena) collected by a second Cimel CE-318 sun/sky photometer within the AERONET network. Ispra exhibited a $SAE^c/AAE^c$ matrix very similar to that observed in Modena (not shown). The resulting $AAE^c$ values for BC, BrC and dust in Modena and Ispra (Table 1) are consistent

**Table 1.** Summary table of columnar Absorption Angstrom Exponent (AAE[c]) for BC, BrC and dust by this work and other literature studies. Rows are organised by wavelength.

| Citation | Setting | BC or alike Wavelength (nm) | BC or alike AAE[c] | BrC or alike Wavelength (nm) | BrC or alike AAE[c] | Dust Wavelength (nm) | Dust AAE[c] |
|---|---|---|---|---|---|---|---|
| This work | AERONET, Modena, Italy | 440 – 675 | 1.12 ± 0.11 | 440 – 675 | 4.35 ± 1.28 | 440 – 675 | 2.83 ± 0.69 |
| This work | AERONET, Ispra, Italy | 440 – 675 | 1.11 ± 0.10 | 440 – 675 | 4.33 ± 1.04 | 440 – 675 | 3.51 ± 0.97 |
| Bahadur et al. (2012) | AERONET, worldwide | 440 – 675 | 0.55 ± 0.24 | 440 – 675 | 4.55 ± 2.01 | 440 – 675 | 2.20 ± 0.50 |
| This work | AERONET, Modena, Italy | 675 – 870 | 1.10 ± 0.11 | 675 – 870 | – | 675 – 870 | 1.06 ± 0.57 |
| This work | AERONET, Ispra, Italy | 675 – 870 | 1.13 ± 0.10 | 675 – 870 | – | 675 – 870 | 0.97 ± 0.56 |
| Bahadur et al. (2012) | AERONET, worldwide | 675 – 870 | 0.85 ± 0.40 | 675 – 870 | – | 675 – 870 | 1.15 ± 0.50 |
| Dubovik et al. (2002) | AERONET, worldwide | 440 – 870 | 0.4 – 2.5[a] | | | 440 – 870 | 0 – 1.6 |
| Giles et al. (2012) | AERONET, worldwide | 440 – 870 | 1.0 – 1.4 | | | 440 – 870 | 1.5 – 2.3 |
| Russell et al. (2010) | AERONET, worldwide | 440 – 870 | ∼ 0.7 – 1.2[a] | | | 440 – 870 | ∼ 1.5 – 2.6 |
| Zhang et al. (2022) | AERONET/GRASP[b], worldwide | 440 – 870 | 1.1 – 1.2[a] | | | 440 – 870 | ∼ 1.2 – 3 |
| Mallet et al. (2013) | AERONET, Mediterrenean | | | | | 440 – 870 | ∼ 1.96[c] |
| Kayetha et al. (2022) | OMI-MODIS-AERONET, worldwide | 340 – 646 | 1.0 – 1.3[a] | | | 340 – 646 | 2.7 – 3.8 |
| Zhu et al. (2021) | SKYNET[d], Fukue, Japan | | | 340 – 500 | 5.3 | | |

[a] These values are referred generically to 'urban/industrial/polluted aerosol', i.e. potentially from a mixture of BC and BrC

[b] GRASP: Generalized Retrieval of Aerosol and Surface Properties (Dubovik et al., 2014)

[c] These values are referred generically to 'dusty sites'

[d] SKYNET is a worldwide network of sun/sky photometers (Nakajima et al., 2020)

with most of the existing literature and the variability reported therein (e.g. Russell et al., 2010; Bahadur et al., 2012; Giles et al., 2012; Kayetha et al., 2022).

Finally, with reasonable confidence in the tailored AAE values for the different absorbing components, each AERONET retrieval at Modena and Ispra was apportioned by summarizing the solutions of the equation system as described by Bahadur et al. (2012). The apportionment was performed by the following two step procedure, based on the assumption that AAE[c] followed a normal distribution featured by the parameters in Table 1.

– Step 1. random extraction of AAE[c] for all species at all wavelengths

– Step 2. direct solution of the system of Ångström equations

The steps 1 and 2 were repeated $10^4$ times for each retrieval in order to develop statistics of the AAE[c] combination that provides a solution to the system. The time series of median AAE[c] values was fairly stable over the measurement period, at both sites, except during an intense episode of dust transport, when the AAE[c] for BrC increased significantly and AAE2[c] for dust dropped (Figure S3). Both AAE[c] for BrC and AAE2[c] for dust values were on the tails of their respective distributions. It is worth noting that AAE[c] refers to BC and not to eBC since it proceeds from a direct estimate of the absorption wavelength dependence of aerosol particles while suspended in air.

 **3   Results**

**3.1   Diurnal patterns for the in-situ data**

Figure 2 shows the medians and interquartile ranges of atmospheric species obtained from in-situ observations along with hourly traffic count from the induction loops closest to each monitoring site for winter (December, January and February) from early 2020 until March 2021. This data is screened for days with non-negligible dust load, as specified in 2.4.1. The $\sigma_{ap}$ at 528
nm for winter weekdays (Monday through Friday) and winter holidays (i.e. Sundays, local and national holidays) is in the top panel of Figure 2 and represents the absorption by aerosol at about 4 m above the ground. Saturdays are excluded due to their mixed signal between a holiday and a weekday. The pattern of absorption apportionment components at 880 nm is also shown, followed by NO, $NO_2$, $O_3$ (UB only) and $C_6H_6$ (UT only). Figure 3, based on the same dataset as Figure 2, displays the share of $\sigma_{ap}$ at 375 nm due to BC from fossil fuel, BC from biomass burning and to BrC, along with the variability in AAE over
the range 375 – 880 nm. The medians and interquartile ranges for meteorological variables over the same period are shown in Figure 4, while the hourly wind rose is shown in Figure S4. Overall median and interquartile ranges for the dataset in Figure 2 (i.e. with dust screening) and Figure 4 are shown in Tables 2 and S1 respectively.

Table 3 reports a comparison of aerosol and BrC absorption reported by this and other studies at a few Po valley sites. Values in Modena are generally higher than those reported in urban background Milan and the rural background sites of Motta
Visconti and Ispra (Ferrero et al., 2021b; Gilardoni et al., 2020a; Laj et al., 2020; Zanatta et al., 2016). The data from these earlier studies comes from filter absorption photometers, either MAAP or aethalometers, with the latter instrument corrected for multiple-scattering-induced bias based on co-located observations. For example in Gilardoni et al. (2020a) a $C_{ref}$ = 3.0 based on Collaud Coen et al. (2010) was used to correct AE22 absorption. Compared to other Southern European urban sites, Modena recorded larger $\sigma_{ap}$ at 660 nm than Barcelona (Ealo et al., 2018), but lower $\sigma_{ap}$ at 375 nm than Athens (Greece),
mainly due to the large impact by biomass burning emissions in this city (Liakakou et al., 2020; Katsanos et al., 2019). A similar pattern is observed for BrC absorption. No MAAP co-location was available for the two MA200s in Modena, leading to a larger uncertainty in their absolute readings; however these two units showed good agreement with a MAAP during a BC intercomparison in urban background Athens (Stavroulas et al., 2022), where they exhibited a linear slope of 1.00 ($R^2$ = 0.92) in winter and 1.07 ($R^2$ = 0.92) in summer.

The diurnal pattern for absolute levels of fossil fuel combustion species (Figure 2) exhibits a similar pattern. There is an initial increase during the morning rush hour (8:00 to 10:00 Local Time, LT), followed by a drop at midday due to the dilution induced by an increased MLH depth despite the steady traffic rate. A second increase occurs at 18:00 LT followed by a drop at approximately 20:00 LT, delayed compared to the drop in traffic, possibly because of the shallow MLH in the evening or because of frequent thermal inversions at ground level. More specifically, on weekday evenings $\sigma_{ap}^{BC,ff}$ peaks at 20:00 LT, one
hour later than on holidays, at both UB and UT, with the former site recording $\sigma_{ap}^{BC,ff}$ levels higher in the evening than in the morning.

This main pattern is followed by all atmospheric species except for secondary pollutants (e.g. $O_3$) and aerosols related to biomass burning (e.g. $\sigma_{ap}^{BC,bb}$ and $\sigma_{ap}^{BrC}$). The pattern features higher concentrations during weekdays than holidays at both sites,

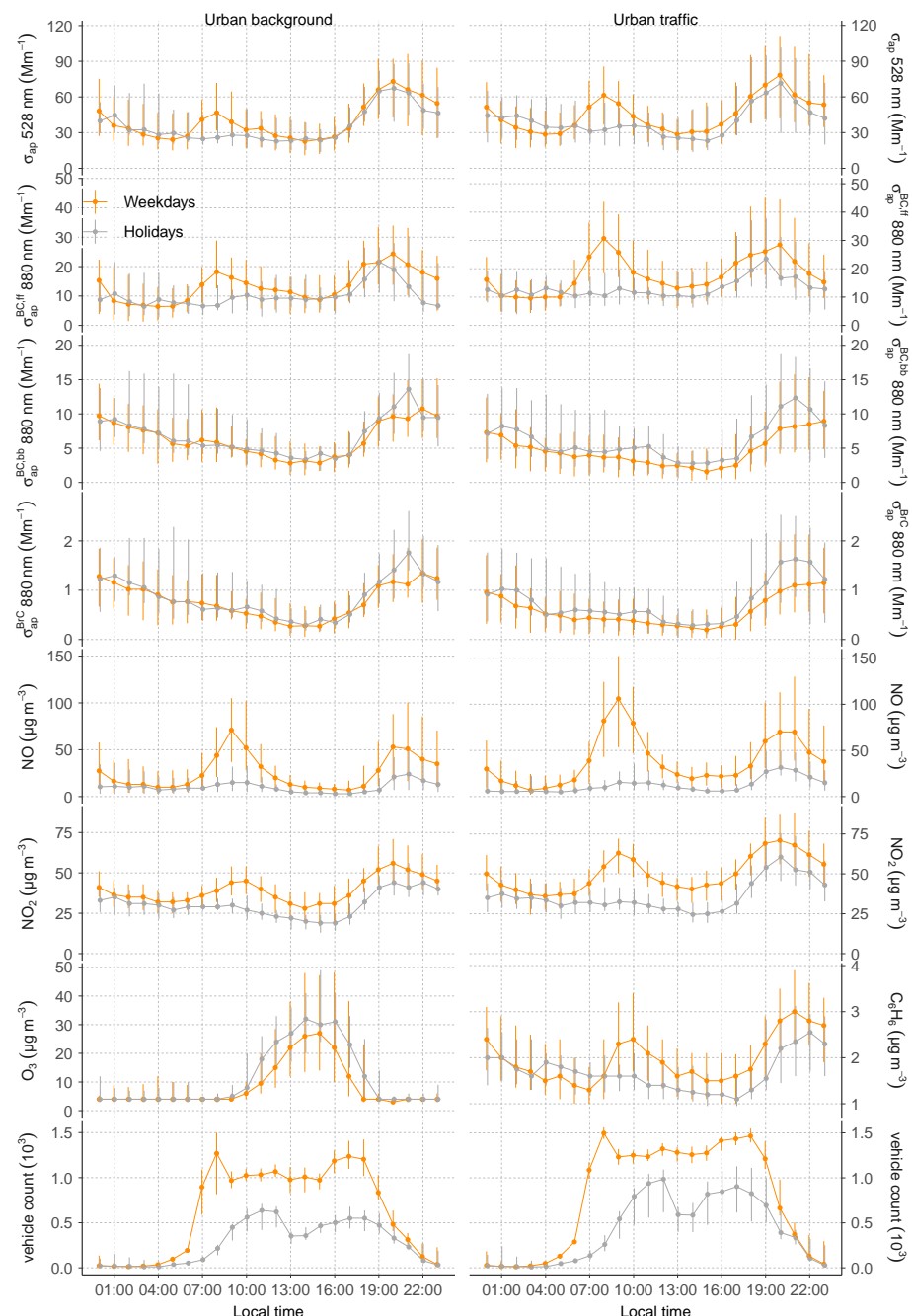

**Figure 2.** Diurnal pattern for the medians and the interquartile ranges for total $\sigma_{ap}$ at 528 nm, the apportioned $\sigma_{ap}$ at 880 and regulatory gas compounds at the urban background (left) and urban traffic (right) air quality monitoring site, for weekdays (orange) and holidays (grey).

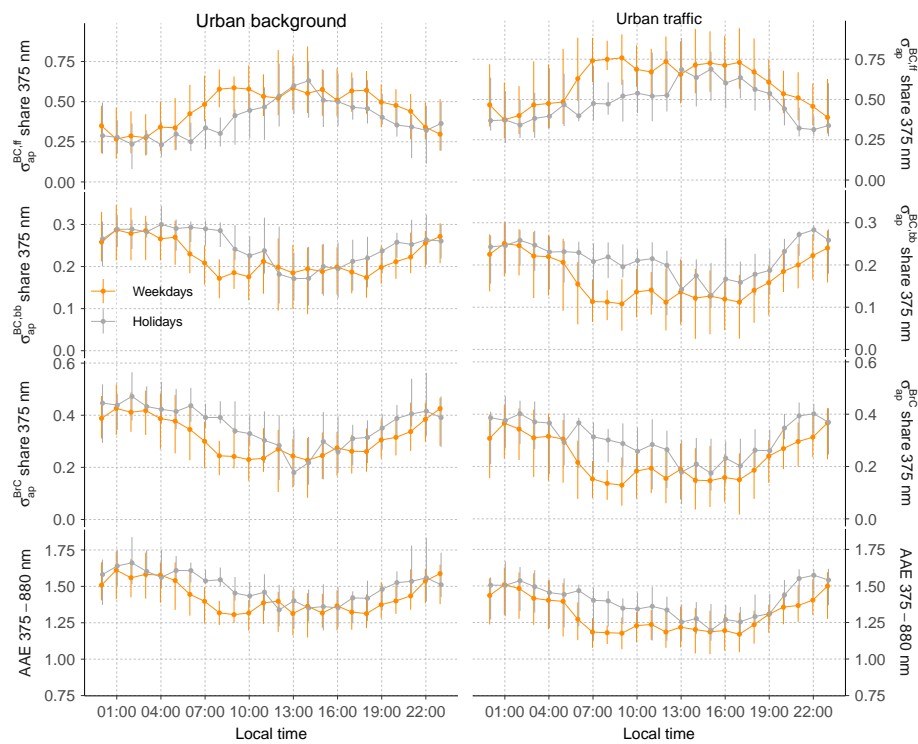

**Figure 3.** Diurnal pattern for the medians and the interquartile ranges for apportioned $\sigma_{ap}$ at 375 nm and of the AAE (375 – 880 nm) at the urban background (left) and urban traffic (right) air quality monitoring site, for weekdays (orange) and holidays (grey).

**Table 2.** Summary table of atmospheric species and traffic volume for the UB and the UT site.

| Variable | Urban background | | | | | | Urban traffic | | | | | |
|---|---|---|---|---|---|---|---|---|---|---|---|---|
| | Weekdays | | | Holidays | | | Weekdays | | | Holidays | | |
| | med | 25th q | 75th q | med | 25th q | 75th q | med | 25th q | 75th q | med | 25th q | 75th q |
| $\sigma_{ap}$ 528 nm (Mm$^{-1}$) | 37.1 | 20.5 | 63.0 | 32.3 | 21.4 | 63.9 | 42.4 | 24.8 | 69.1 | 38.1 | 21.6 | 62.7 |
| $\sigma_{ap}^{BC,ff}$ 880 nm (Mm$^{-1}$) | 12.6 | 5.5 | 22.2 | 9.6 | 4.6 | 19.0 | 16.9 | 9.1 | 28.1 | 12.7 | 7.0 | 20.6 |
| $\sigma_{ap}^{BC,\,bb}$ 880 nm (Mm$^{-1}$) | 6.1 | 3.1 | 10.6 | 6.3 | 3.7 | 12.0 | 4.3 | 1.4 | 8.8 | 5.1 | 2.3 | 11.1 |
| $\sigma_{ap}^{BrC}$ 880 nm (Mm$^{-1}$) | 0.8 | 0.3 | 1.4 | 0.8 | 0.4 | 1.5 | 0.5 | 0.1 | 1.1 | 0.7 | 0.2 | 1.5 |
| NO (µg m$^{-3}$) | 20 | 6 | 49 | 8 | 4 | 21 | 31 | 12 | 64 | 10 | 5 | 25 |
| NO$_2$ (µg m$^{-3}$) | 39 | 29 | 48 | 29 | 21 | 38 | 48 | 37 | 61 | 34 | 25 | 46 |
| O$_3$ (µg m$^{-3}$) | 5 | 4 | 16 | 7 | 4 | 24 | – | – | – | – | – | – |
| C$_6$H$_6$ (µg m$^{-3}$) | – | – | – | – | – | – | 1.9 | 1.3 | 2.7 | 1.6 | 1.1 | 2.3 |
| vehicles per day ($10^3$) | 16.1 | 14.3 | 31.9 | 7.2 | 5.9 | 9.1 | 19.9 | 17.2 | 38.0 | 11.3 | 7.7 | 13.9 |

as shown by the 7% – 240% increase on weekdays, similar to the 230% – 250% increase in traffic, confirming a major and
local fossil fuel combustion direct origin. For most of these species the absolute interquartile range (IQR) is larger on weekdays

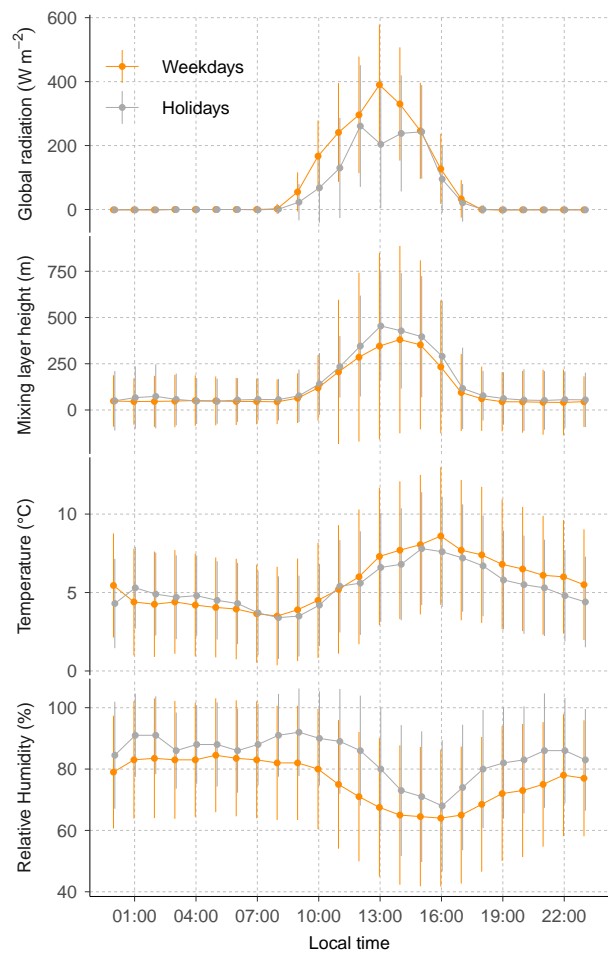

**Figure 4.** Diurnal pattern for the medians and the interquartile ranges of meteorological variables in winter (DJF) during weekdays and holidays All variables proceed from an urban meteorological station, besides mixing layer height, provided by ERA5 reanalysis.

than holidays. For gas phase compounds the IQR increased on weekdays from 8% for $NO_2$ at UT to 160% for NO at the UB. For absorption the largest increase in IQR (a 56% increase) occurred for $\sigma_{ap}^{BC,ff}$ at the UT. The difference in variability between weekdays and holidays might be partly driven by the larger variability in fossil fuel combustion emissions during the former, along with the larger count of weekdays compared to holidays in the statistics.

$\sigma_{ap}^{BC,bb}$ and $\sigma_{ap}^{BrC}$ exhibit a diurnal pattern featuring an increase from 17:00 LT to 23:00 LT possibly triggered by the decrease in both the MLH and $T$, leading to condensation of semivolatile organics and to an increase in biomass burning emissions. A similar diurnal pattern for BrC absorption was observed in UB Milan and rural background Po valley (Gilardoni et al., 2020a), and in UB Athens (Liakakou et al., 2020; Kaskaoutis et al., 2021). The weekly pattern for these two species is larger at the UT, with an increase during holidays in the overall median values of $\sigma_{ap}^{BC,bb}$ and $\sigma_{ap}^{BrC}$ of 22% and 35% respectively, along with an increase in their IQR of 16% and 28%.

**Table 3.** Summary table of mean ± standard deviation of absorption for aerosol and BrC based on this work and other literature studies. Rows are organised by wavelength. UB, UT and RB stand for Urban Background, Urban Traffic and Rural Background respectively.

| Citation | Setting | Period | Wavelength (nm) | Absorption (Mm$^{-1}$) |
|---|---|---|---|---|
| | | | | Aerosol |
| This work | MA200, UB, Modena, Italy | winter 2019 – 2021 | 880 | 22.1 ± 15.5 |
| This work | MA200, UT, Modena, Italy | winter 2019 – 2021 | 880 | 27.3 ± 21.0 |
| Gilardoni et al. (2020a) | AE22, UB, Milan, Italy | winter 2015 – 2016 | 880 | 12.1 ± 8.5 |
| Gilardoni et al. (2020a) | AE22, RB, Motta Visconti, Italy | winter 2015 – 2016 | 880 | 7.6 ± 7.1 |
| Ferrero et al. (2021b) | AE31, UB, Milan, Italy | December 2015 | 880 | 31.1 ± 0.5[a] |
| This work | MA200, UB, Modena, Italy | winter 2019 – 2021 | 375 | 75.4 ± 52.2 |
| This work | MA200, UT, Modena, Italy | winter 2019 – 2021 | 375 | 84.7 ± 67.7 |
| Gilardoni et al. (2020a) | AE22, UB, Milan, Italy | winter 2015 – 2016 | 370 | 38.8 ± 27.6 |
| Gilardoni et al. (2020a) | AE22, RB, Motta Visconti, Italy | winter 2015 – 2016 | 370 | 28.7 ± 30.1 |
| Kaskaoutis et al. (2021) | AE33, UB, Athens, Greece | winter 2016 – 2017 | 370 | 82.8 ± 133.3 |
| This work | MA200, UB, Modena, Italy | winter 2019 – 2021 | 625 | 25.1 ± 2.4[b] |
| This work | MA200, UT, Modena, Italy | winter 2019 – 2021 | 625 | 30.3 ± 2.4[b] |
| Zanatta et al. (2016) | MAAP, RB, Ispra, Italy | winter 2008 – 2011 | 637 | 18.6 ± 1.7[b] |
| Ealo et al. (2018) | MAAP, UB, Barcelona, Italy | winter 2009 – 2014 | 637 | ~ 17.4 |
| This work | MA200, UB, Modena, Italy | winter 2019 – 2021 | 625 | 28.1 (8.1 – 69.2)[c] |
| This work | MA200, UT, Modena, Italy | winter 2019 – 2021 | 625 | 33.3 (10.0 – 80.3)[c] |
| Laj et al. (2020) | AE31, RB, Ispra, Italy | winter 2016 | 660 | 17.3 (2.5 – 48.2)[c] |
| | | | | BrC |
| This work | MA200, UB, Modena, Italy | winter 2019 – 2021 | 375 | 26.6 ± 22.2 |
| This work | MA200, UT, Modena, Italy | winter 2019 – 2021 | 375 | 23.9 ± 24.2 |
| Gilardoni et al. (2020a) | AE22, UB, Milan, Italy | winter 2015 – 2016 | 370 | 6.0 ± 2.7[d] |
| Gilardoni et al. (2020a) | AE22, RB, Motta Visconti, Italy | winter 2015 – 2016 | 370 | 5.3 ± 3.0[d] |
| Kaskaoutis et al. (2021) | AE33, UB, Athens, Greece | winter 2016 – 2017 | 370 | 36.7 ± 73.6 |

[a] 95% confidence interval of the mean

[b] geometric mean ± geometric standard deviation

[c] median (10th - 90th quantile)

[d] BrC determined on methanol extraction

The share of absorption at 375 nm for the three apportioned species (Figure 3) shows a distinct diurnal and weekly pattern at the UT, with $\sigma_{ap}^{BC,bb}$ and $\sigma_{ap}^{BrC}$ being ca. 37% larger during holidays, in contrast to $\sigma_{ap}^{BC,ff}$ which is 32% larger on weekdays. The UB exhibited a similar pattern, although with lower intensity. The holiday increase in biomass burning aerosol is probably linked to the longer stay at home compared to weekdays and to a large recreational use of biomass burning in town, where most houses use compressed natural gas for domestic heating and cooking (99.4% of $PM_{10}$ emissions by SNAP 2 in Modena are from biomass combustion for domestic heating according to ARPAE (2020)). The diurnal pattern of the share of absorption

by $\sigma_{\mathrm{ap}}^{\mathrm{BC,ff}}$ at 375 nm is similar to the diurnal traffic count cycle, exhibiting larger values during weekdays and at the UT. This supports the results of the apportionment and the hypothesis of the role of the MLH in the evening enhancement of absorption.

$O_3$ exhibits a 'weekend effect' (Cleveland et al., 1974), common to most urban areas in Europe having a VOC–limited regime, i.e. on holidays ozone rises earlier in the morning due to the lower $NO_x$ levels, leading to a more efficient photocatalytic cycle, and drops later in the evening due to the (later) increase in $NO_x$.

Atmospheric heating by aerosols based on $\sigma_{\mathrm{ap}}$ values in Modena was estimated by determining the HR from AERONET data as detailed in section 2.2. Figure 5a shows the complete HR time series obtained over Modena during the investigated period. Under an average (standard deviation) irradiance value of 386 (143) $\mathrm{W\,m^{-2}}$ the average (standard deviation) HR was 1.61 (1.58) $\mathrm{K\,d^{-1}}$. This value is consistent with data from Milan for wintertime, under clear sky conditions, where a mean (± mean confidence interval) of 1.68 ± 0.04 $\mathrm{K\,d^{-1}}$ was found, when the incoming radiation was similar (441 ± 148 $\mathrm{W\,m^{-2}}$). This latter is an important point since AERONET data is mainly available under clear sky conditions and thus the obtained HR data represents the upper limit for the site. In the Po valley the HR was shown to decrease by a ∼12% factor for every okta of sky covered by clouds (Ferrero et al., 2021b). With respect to the HR diurnal pattern, Figure 5b shows the mean diurnal pattern of irradiance and HR under clear sky and cloudy conditions. The incoming radiation peaked at 13:00 LT with 529 ± 55 $\mathrm{W\,m^{-2}}$ (Figure 5b) while $\sigma_{\mathrm{ap}}$ peaked between 8:00 and 10:00 LT (Figure 2). This causes an asymmetric HR diurnal pattern, characterized by a fast increase to the maximum at 11:00 LT (1.83 ± 0.84 $\mathrm{K\,d^{-1}}$) and a subsequent slower decrease till sunset (Figure 5b), as is common under clear sky conditions (Ferrero et al., 2018, 2021b).

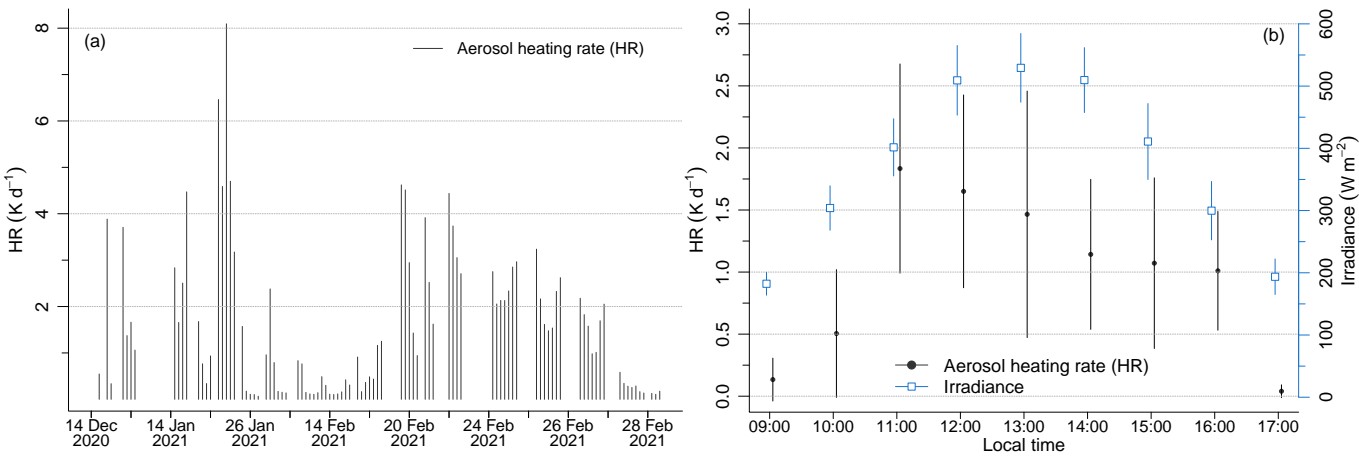

**Figure 5.** Time series (a) and diurnal pattern (b) of aerosol heating rate (HR) at Modena by AERONET retrievals.

The wind pattern in Modena features a mild mountain-valley breeze system along the Po valley longitudinal axis, superimposed on the local wind circulation. During the investigated winter period, calm wind conditions (speed lower than 1 $\mathrm{m\,s^{-1}}$) occurred 25% of the time and an overall wind speed average of ca. 1.5 $\mathrm{m\,s^{-1}}$ was recorded. NW winds, blowing from the higher side of the valley, dominated during daytime hours (11:00 - 17:00 LT) and were associated with the highest windspeed

(occasionally above 9 m s$^{-1}$). The rest of the day features local W-SW low winds (windspeeds lower than 3 m s$^{-1}$) and some easterly winds (Figure S4).

A conditional bivariate polar function was applied to NO$_x$, $\sigma_{ap}^{BC,ff}$ at 880 nm and $\sigma_{ap}^{BrC}$ at 375 nm at both the UT and UB sites, to identify the position of potential emission sources (Figure 6) on weekdays and holidays, excluding days with significant dust load at ground. Wind speed and direction data were combined with atmospheric compounds levels by the use of conditional bivariate polar functions (Uria-Tellaetxe and Carslaw, 2014), as implemented in the R-software package Openair (Carslaw and Ropkins, 2012). This tool provides information on both the direction and the distance of the (relatively local) emission sources

which are contributing significantly to the observed concentration levels, as well as on the wind direction sectors which might provide clean air masses.

At the UB, NO$_x$ and absorption exhibit slightly different directional patterns: both show an increase associated with slow S–SW winds, particularly on weekdays, while $\sigma_{ap}^{BC,ff}$ exhibits an increase on weekdays during NE moderate winds, probably linked to traffic on the busy road 400 m in that direction. $\sigma_{ap}^{BrC}$ exhibits larger values during holidays during southerly winds,

which is different than the pattern for BC from fossil fuels and NO$_x$ for the same period. At the UT site the directional pattern between NO$_x$ and $\sigma_{ap}^{BC,ff}$ are quite similar, highlighting the role of nearby traffic during weekdays and of the major east–west road south of the UT site which contributes mainly during holidays. Also at the UT $\sigma_{ap}^{BrC}$ is higher during holidays and under southerly winds. This latter increase occurs during evening/night hours (not shown), consistent with biomass burning from domestic heating for recreational use, with the increase probably enhanced by nighttime atmospheric stagnation. Finally, NW

moderate winds are associated with low levels in NO$_x$, $\sigma_{ap}^{BC,ff}$ and $\sigma_{ap}^{BrC}$, mainly because NW winds occur primarily at midday during maximum atmospheric mixing.

### 3.2    Comparing absorption optical depth from remote sensing and in situ values

In-situ and columnar aerosol optical properties were compared to assess both how representative the surface in-situ aerosol optical measurements are of the mixed layer, and how the absorption within the MLH compares to the atmospheric column.

Urban in-situ and column data was compared over the whole time period, although simultaneous observations were mainly available only in February 2021. For this comparison the in-situ $\sigma_{ap}$ were rescaled over (i.e. multiplied by) the MLH height, resulting in an estimate of the integral aerosol absorption over the MLH height representing the case of vertically homogenous $\sigma_{ap}$ from the ground to the top of this atmospheric layer. The in-situ $\sigma_{ap}$ in the IR spectral range ($\lambda$ = 880 nm) rescaled over the MLH height was generally larger than AAOD (Figure 7), for both the L1.5* and L2.0 AERONET inversions, with mean

normalised errors of MNE = 2.2 and MNE = 1.7 respectively. A better agreement occurred for blue wavelengths ($\lambda$ = 470 nm and 440 nm for the in-situ and the columnar observations respectively) with MNE = 1.0 and MNE = 0.8 for L1.5* and L2.0 respectively. These results suggests an inhomogeneous vertical distribution of aerosols, i.e. most likely the occurrence of a very large accumulation of aerosols at the ground layer if compared to the atmospheric column and to the MLH, similar to previous observations in Milan during very stable atmospheric conditions (Ferrero et al., 2011). The overestimation of scaled

in-situ aerosol properties compared to columnar aerosol properties observed in this study may be affected by some concurrent conditions: (a) the large role of traffic and of other ground emissions on aerosol absorption (b) a persistent ground thermal

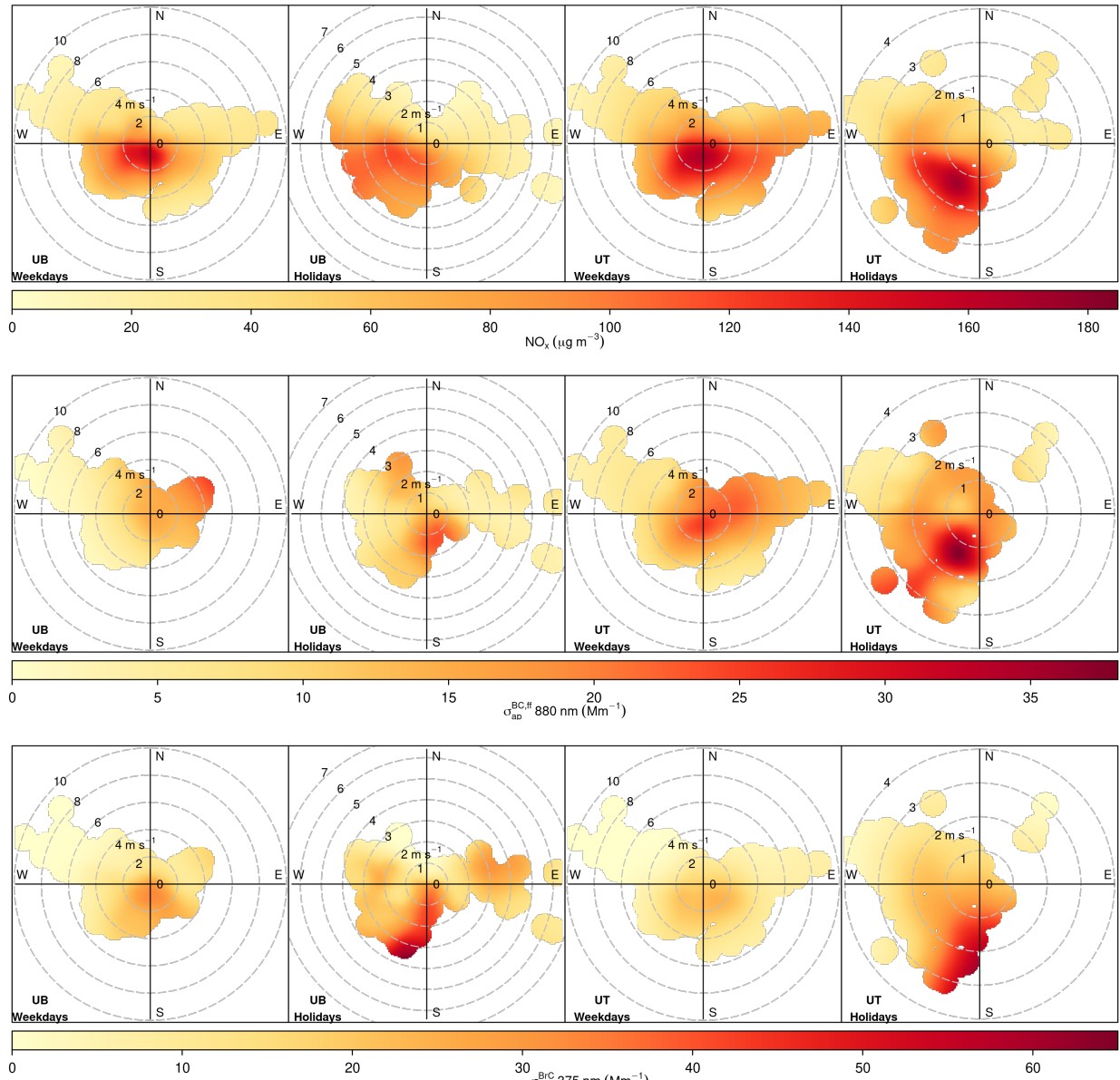

**Figure 6.** Bivariate polar function applied to $NO_x$ (first row), $\sigma_{ap}^{BC,ff}$ at 880 nm (second row), $\sigma_{ap}^{BrC}$ at 375 nm (third row) at the UB and UT sites, for weekdays and holidays.

inversion occasionally as low as few hundred meters, according to radiosoundings at 12 UTC at the rural Po valley site of San Pietro Capofiume (c) a bias in the ERA5 estimate of the MLH. These conditions mainly contribute to the significantly larger values observed in scaled ground absorption, particularly at 880 nm, where fossil fuel emissions provide the largest

contribution. At remote sites an opposite pattern was consistently found (e.g. Bergin et al., 2000; Slater and Dibb, 2004; Aryal

et al., 2014; Chauvigné et al., 2016), where MLH-scaled surface in-situ atmospheric extinction underestimates sun photometry observations of AOD, mainly because of aerosol hygroscopicity (in-situ measurements are typically made at low RH) and aerosol layers above the MLH.

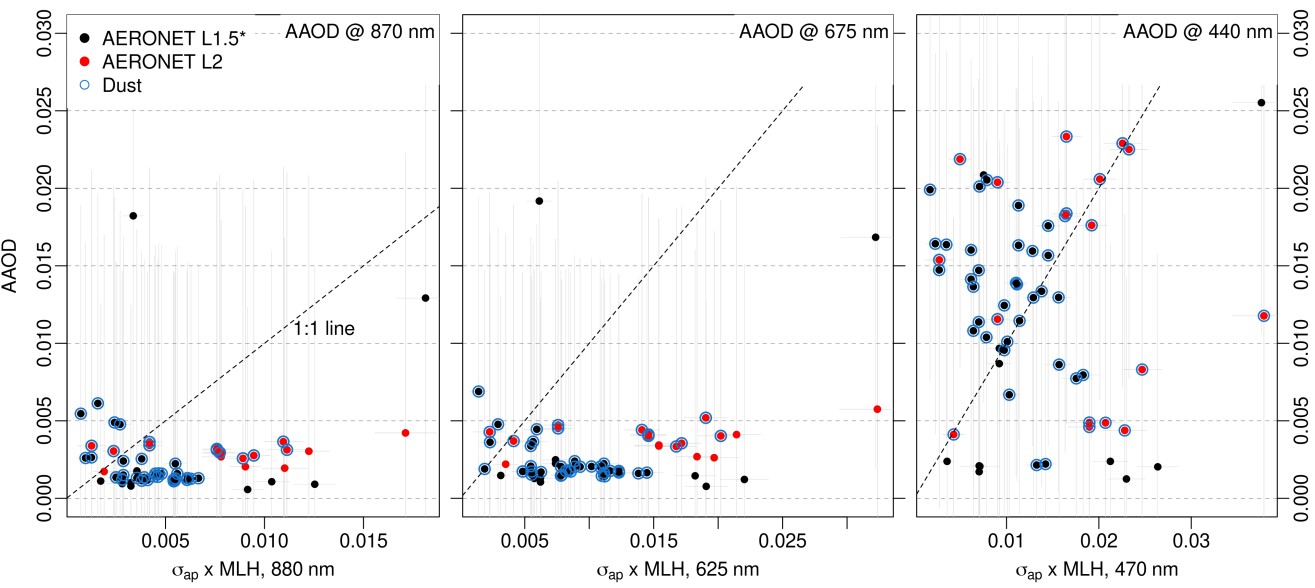

**Figure 7.** Absorption aerosol optical depth based on columnar and in-situ observations in the IR (left), Green (centre) and UV (right) regions, using both L1.5 and L2.0 AERONET retrievals. Bars indicate measurement uncertainty.

For an assessment of the role of the MLH and of atmospheric layers in the discrepancies between surface and column observations mentioned above, an analysis of the Apparent Aerosol Optical Height (ApAOH) was performed. ApAOH can be defined as the ratio between AAOD and $\sigma_{ap}$ (giving ApAOH units of length) to represent the atmospheric depth below which aerosols are uniformly distributed (Loía-Salazar et al., 2014). In the case of well-mixed conditions for absorbing aerosols, ApAOH is similar to the MLH, while larger differences indicate less vertical mixing of the aerosol particles.

The comparison of ApAOH using L1.5* and MLH in Figure 8a shows how ApAOH is, on average, lower than the MLH, at both 880 (870) nm (Mean Error ME = −243 m) and 470 (440) nm (ME = −96 m) wavelengths of the in-situ (columnar) instruments. In the period February 22nd - 26th ApAOH was highly consistent with the MLH for 470 (440) nm showing a ME = 31 m, i.e. similar to the bias reported for ERA5 estimates of MLH in Europe by Guo et al. (2021). Conversely ApAOH at 880 (870) nm was significantly lower than MLH (ME = −190 m), suggesting a different vertical mixing between two absorbing aerosol species. The end of February 2021 featured the development of a strong anticyclone system in the Mediterranean basin, leading to above-average atmospheric temperature in Southern and Central Europe, clear sky (as shown by the high frequency of retrievals), the build-up of atmospheric pollutants and the arrival in Italy of Saharan dust rich air masses. During the development of this high pressure system, daily soundings collected at 00 UTC and 12 UTC at the rural site of San Pietro Capofiume (44.65° N, 11.62° E, 60 km east of Modena) show the progressive vertical drop of a thermal

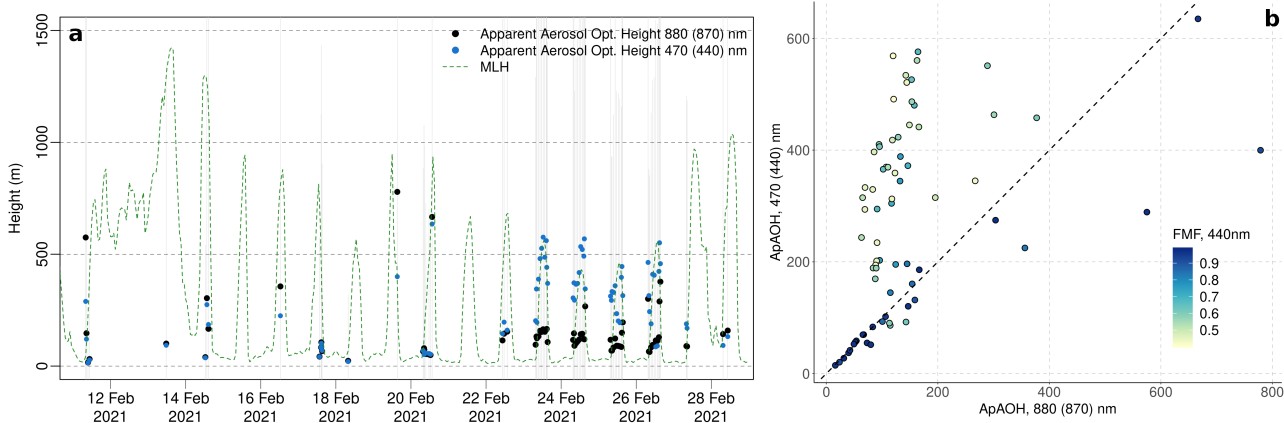

**Figure 8.** a: Apparent aerosol optical height (ApAOH) computed based on aerosol absorption at 880 nm (870 nm) and 470 nm (440 nm) by in-situ (columnar, L1.5*) instruments. Simulated MLH depth by ERA5 is also plotted. b: ApAOH computed based on aerosol absorption at 880 nm (870 nm) and 470 nm (440 nm) by in-situ (columnar) instruments and color-coded according to the Fine Mode Fraction (FMF) at 440 nm. The dashed line indicates the 1:1 line.

inversion from ca. 2km (on Feb 20th) to ca. few hundred meters (on Feb 26th), leading to above seasonal median levels
for the in-situ $\sigma_{\mathrm{ap}}$ at both 880 nm and 470 nm (Figure S6). Concurrent AAOD observations were above the median at 440 nm and within the seasonal median at 870 nm, leading to different ApAOH values for these two wavelengths (Figure 8a). Differences in AAOD might originate from a different atmospheric layering and mixing of aerosols species: the volume size distribution from the AERONET inversion shows a switch from a major modal peak in submicron diameters on February 20th (Figure S7a) to a modal peak in the supermicron diameter range on February 23rd (Figure S7b), which lasted until February
27th at midday. Consistently AOD at 500 nm derived by the AERONET Spectral De-Convolution Algorithm (Sinyuk et al., 2020) had a monthly minimum during this clear sky period, showing a switch from a fine aerosol controlled AOD (until Feb 20th) to a coarse aerosol controlled AOD (since Feb 23rd). This suggests that ApAOH at 470 (440) nm is similar to the MLH depth most likely because of a good vertical mixing of dust aerosol, as shown by the decrease in $\mathrm{PM}_{2.5}$ / $\mathrm{PM}_{10}$ ratio over the same period (Figure S1), while the low ApAOH in the IR is probably due to the dominant contribution of (ground level)
traffic emissions to the $\sigma_{\mathrm{ap}}$ at this latter wavelength. This suggests that, in this case, the radiative effect by traffic emissions was relevant mainly at the urban scale. Figure 8b shows how ApAOH at 880 (870) nm and 470 (440) nm are correlated and mainly lay on the 1:1 line during high $\mathrm{FMF}_{440}$ conditions and aerosol volume size distribution with a fine mode peak (Figure S7a). For $\mathrm{FMF}_{440} > 0.8$, linear Pearson's correlation $r = 0.91$, with $\mathrm{FMF}_{440}$ indicating the contribution by fine aerosol to AOD in the Blue range, where dust is a significant absorber. During low $\mathrm{FMF}_{440}$ the ApAOH at 470 (440) nm increases significantly, in
contrast to ApAOH at 880 (870), nonetheless the correlation between the two remains.

### 3.3 Comparison of the contribution by BrC to absorption based on in-situ and columnar data

The contribution of BrC to absorption in Modena according to the in-situ and to the columnar L1.5* data was also compared (Figure 9). Days with significant dust load were removed from the comparison, because the MWAA apportionment method does not include dust absorption. This is necessary since, as shown by the ApAOH, the vertical mixing of dust can be significant, affecting both columnar and in-situ observations. Figure 9 compares the contribution of biomass burning to absorption in Blue (470 nm and 440 nm for the columnar and in-situ observations respectively) by the two apportionment models for 17 matched data points. Calculated statistics indicate a $ME = 0.04$, a $MNE = 1.23$ and low linear correlation (Pearson's $r = 0.39$, Spearman's $\rho = 0.34$). According to columnar retrievals the biomass burning contribution ranged between $2\%$ and $23\%$, with a median (median absolute deviation) of $3.9\%$ ($0.8\%$) and a mean (standard deviation) of $5.3\%$ ($4.3\%$); in-situ observations exhibited a similar range ($0\% - 24\%$), but suggested a higher contribution of biomass related to lower MLH depth and a median (median absolute deviation) of $8.7\%$ ($8.7\%$). Despite the uncertainty associated with these estimates, these results highlight how urban BrC emissions have a large impact on the lower levels of the atmosphere, similar to the findings by Ferrero et al. (2011) for BC. This is consistent with the dynamics of biomass burning emissions from domestic heating, featuring a low exit velocity and negligible plume rise, particularly for natural convection fireplaces or traditional wood-stoves. This BrC absorption contribution is based on days with negligible dust content and thus represents a higher end estimate; nonetheless it is lower than values found for polluted urban sites in eastern Asia, e.g. Beijing, Hong Kong, Seoul, and Osaka, where the share of AAOD due to BrC ranged between $12\% - 14\%$ in the UV during non-dust days on a yearly basis (Cho et al., 2019). Even larger contributions than those found for the eastern Asian sites were reported for Europe during winter where a mean $21\%$ of AAOD in Blue by BrC was reported by Wang et al. (2016b), based on 10 years (2005 – 2014) of AERONET data, also excluding 'dust days'. It is worth noting that over the same decade, Wang et al. (2016b) also reported the in-situ mean share of absorption by BrC in winter at Ispra and SIRTA (Paris, France) to be $\sim 23\%$. A similar study in California found that the contribution by BC and BrC to absorption by in-situ and ground based sun/sky photometer can be similar, depending on the vertical mixing of the planetary boundary layer (Chen et al., 2019). They showed according to both methods, the share of BrC absorption at 440 nm was approximately $30\%$. Similarly, during high pollution events in the Kathmandu valley, Kim et al. (2021) found a good correlation between the in-situ and the columnar estimate of BC and BrC. They estimated a similar share of UV absorption by BrC: $34\%$ and $31\%$ by Aethalometer AE33 and AERONET, respectively and a good correlation between the two techniques ($R^2 = 0.71$).

### 3.4 Spatial variability of ground-based columnar retrievals

In addition to checking the vertical mixing of aerosol, spatial mixing in ground-based columnar properties across part of the Po valley during the studied period was also investigated by comparing AERONET L1.5* version 3 retrievals in Modena and Ispra. The observations from the two instruments were matched when collected within the same hour. The AOD data are partly scattered along the 1:1 line (Figure 10), with a significant (at the 95%) Pearson's (Spearman's) correlation coefficient ranging from $r = 0.67$ ($\rho = 0.56$) at 440 nm, to $r = 0.84$ ($\rho = 0.79$) at 1020 nm and an orthogonal regression coefficient ranging between

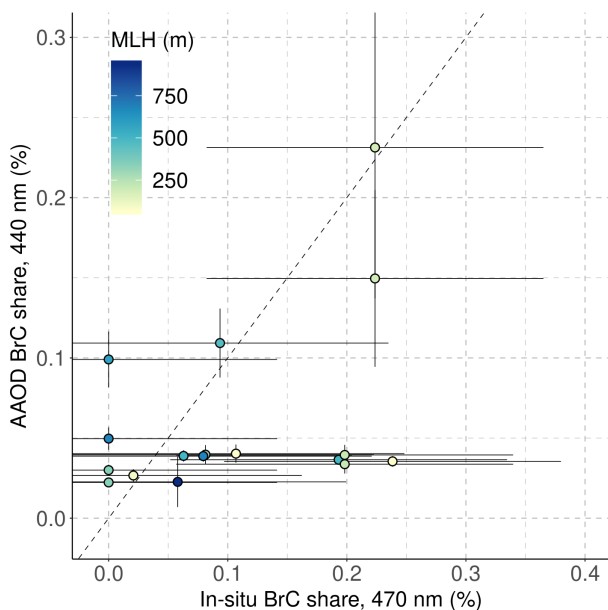

**Figure 9.** Scatter plot of the share of total AAOD due to BrC and the share total in-situ absorption due to BrC. Bars indicate measurement uncertainty.

0.74 at 440 nm and 1.4 at 1020 nm. The high correlation in the AOD at the two sites is mainly driven by the observations in
February, when both sites experienced a drop in FMF and an increase in AOD because of the dust transport event. This dust event was also detected in Ispra by a ground-based LIDAR within the EARLINET network, whose total attenuated backscatter at 500 and 1064 nm showed an aerosol layer between 1.3 – 2.3 km above the ground during February 23 and 24. The layer subsequently dropped to heights between few hundred meters and 1.8 km on February 25 (no LIDAR data is available for February 26th and 27th). The temporal variability in $AOD_{440}$ and $FMF_{440}$ between the two sites also shows a similar pattern
(Figure 10). AOD in Ispra was similar to Modena for most retrievals, with Ispra and Modena having overall median $AOD_{500}$ values of 0.27 and 0.24 respectively. The two sites differed during mid January, when Modena had median $AOD_{500}$ = 0.21, while Ispra had a median $AOD_{500}$ = 0.12, consistent with Modena being a site more representative of urban pollution than Ispra.

Concurrent AAOD retrievals at two sites have a different correlation pattern than AOD (Figure S8), exhibiting the largest
linear correlation at 440 nm ($r$ = 0.70, $\rho$ = 0.72, both statistically significant at the 95% level). The correlation decreases noticeably with increasing wavelength: for 675 nm ($r$ = 0.42, $\rho$ = 0.43, 95% significance) and 870 nm ($r$ = 0.37, $\rho$ = 0.40, 95% significance), while no significant correlation occurred at 1020 nm. In contrast to AOD, AAOD at Ispra was larger than at Modena, with median $AAOD_{440}$ being 0.025 and 0.011 in Ispra and Modena, respectively. The differences in AAOD at the two sites decreased at longer wavelengths, with median $AAOD_{870}$ being 0.003 and 0.002 in Ispra and Modena respectively.
According to the apportionment model (Figure S5) the larger $AAOD_{440}$ at Ispra is due to a larger impact by dust at this site

compared to Modena, with $AAOD_{440,dust}$ being 0.022 and 0.001 at the two sites respectively. $AAOD_{440,dust}$ was the component with the largest correlation between the two sites ($r = 0.62$, $\rho = 0.60$, 95% significance), followed by BC ($r = 0.50$, $\rho = 0.43$, 95% significance), with similar correlation values for $AAOD_{870,dust}$ and $AAOD_{870,BC}$ for these two species (Figure S5), while correlation for BrC absorption at 440 nm was not significant.

## 4  Conclusions

In the urban area of Modena, a town representative of several urban areas in the Po valley (a pollution hotspot for Europe), a set of Light Absorption Aerosol (LAA) observations at multiple wavelengths were collected, along with meteorological and vehicle traffic data. Aerosol absorption was monitored by two *in-situ* MicroAethalometers (MA200), and by a Cimel CE-318 sun/sky-photometer contributing to the AERONET network. The MA200 instruments were deployed at two locations representative of urban background and urban traffic conditions in Modena, while the Cimel sunphotometer was located in urban background conditions in Modena. In-situ observations, apportioned to fossil fuel and biomass burning, were shown to be largely influenced by ground emissions. The comparison of columnar absorption and *in-situ* absorption rescaled over the mixing-layer exhibited contrasting results, demonstrated by a large difference in the infrared region (mean normalised error, MNE, up to 2.2) but with better agreement in the blue wavelength region (MNE = 0.8), confirming the impact of ground emissions on atmospheric levels of LAA. Under the (reasonable) assumption of the generation of most of the AOD signal within the mixing layer, the heating rate by LAA was estimated in 1.61 K d$^{-1}$. The apportionment of columnar absorption to Black Carbon (BC), Brown Carbon (BrC) and dust, along with the aerosol size distribution by AERONET inversion, highlighted the major role of long-range transported dust in driving the correspondence between the *in-situ* and the columnar absorption at 440 nm, indicating a deeper vertical mixing for dust, in contrast to urban ground-based emissions which are confined to lower heights. This latter result was shown specifically for BrC absorption, whose contribution to *in-situ* absorption resulted in a larger contribution to absorption (up to 23%) and featured wider variability, relative to the columnar retrieval of absorption. The spatial extent of the dust impact was evaluated by the combined analysis of concurrent columnar retrievals in Modena and in Ispra (225 km NW of Modena): the sites showed large agreement in AOD (Pearson's linear correlation coefficient $r = 0.84$ at 1020 nm) and in AAOD at 440 nm ($r = 0.70$), where dust has a significant absorption. Consistently the AAOD apportioned to dust was the species with the largest correlation between the two sites, reaching $r = 0.62$ at 440 nm, supporting the occurrence of significant spatial mixing by the transported dust, along with the vertical mixing.

An improved knowledge of the role by the in-urban emissions of LAA is critical to control local air quality, urban heat island effects and climate forcing and an apportionment of LAA based on their atmospheric levels, as presented here, contributes towards this goal. This study provides important insights on the role of the in-situ absorption monitoring in estimating the actual absorption aloft and whether it can be used for radiative forcing estimates. Moreover the characterization of the intra-urban variation of absorbing aerosol based on different site types contributes in the ambient exposure domain. Towards this latter outcome, a more in depth investigation of the contribution of urban areas to atmospheric LAA can be gained by the application of specific atmospheric dispersion tools, and this represents one of the major study outlooks. More specifically, Lagrangian

**Table A1.** Statistical metrics for the assessment of the agreement between th in-situ and the columnar data ($L^{\mathrm{is}}$ and $L_i^{\mathrm{col}}$ respectively)

| | |
|---|---|
| Mean Error (ME) | $\frac{1}{N}\sum_{i=1}^{N}\left(L_i^{\mathrm{is}} - L_i^{\mathrm{col}}\right)$ |
| Mean Normalised Error (MNE) | $\frac{1}{N}\sum_{i=1}^{N}\frac{\left(L_i^{\mathrm{is}} - L_i^{\mathrm{col}}\right)}{L_i^{\mathrm{col}}}$ |

particle dispersion models would provide information on atmospheric levels across the urban area at a fine spatial resolution, supporting advanced exposure studies, and, further, would give an estimate of the spatial- and time-resolved emission factors for LAA in the urban area.

## Appendix A: AAOD apportionment model

AAOD was apportioned by solving the following equation system

$$\mathrm{AAOD}_\lambda = \sum_i \mathrm{AAOD}_\lambda^i \tag{A1}$$


$$\mathrm{AAOD}_\lambda^i = \mathrm{AAOD}_{\mathrm{ref}}(\lambda/\lambda_{\mathrm{ref}})^{-\mathrm{AAE}_\lambda^i} \tag{A2}$$

with $i$ in BC, Dust and BrC and $\lambda$ in 440 nm, 675 nm and 880 nm (BrC only at 440 nm and 675 nm).

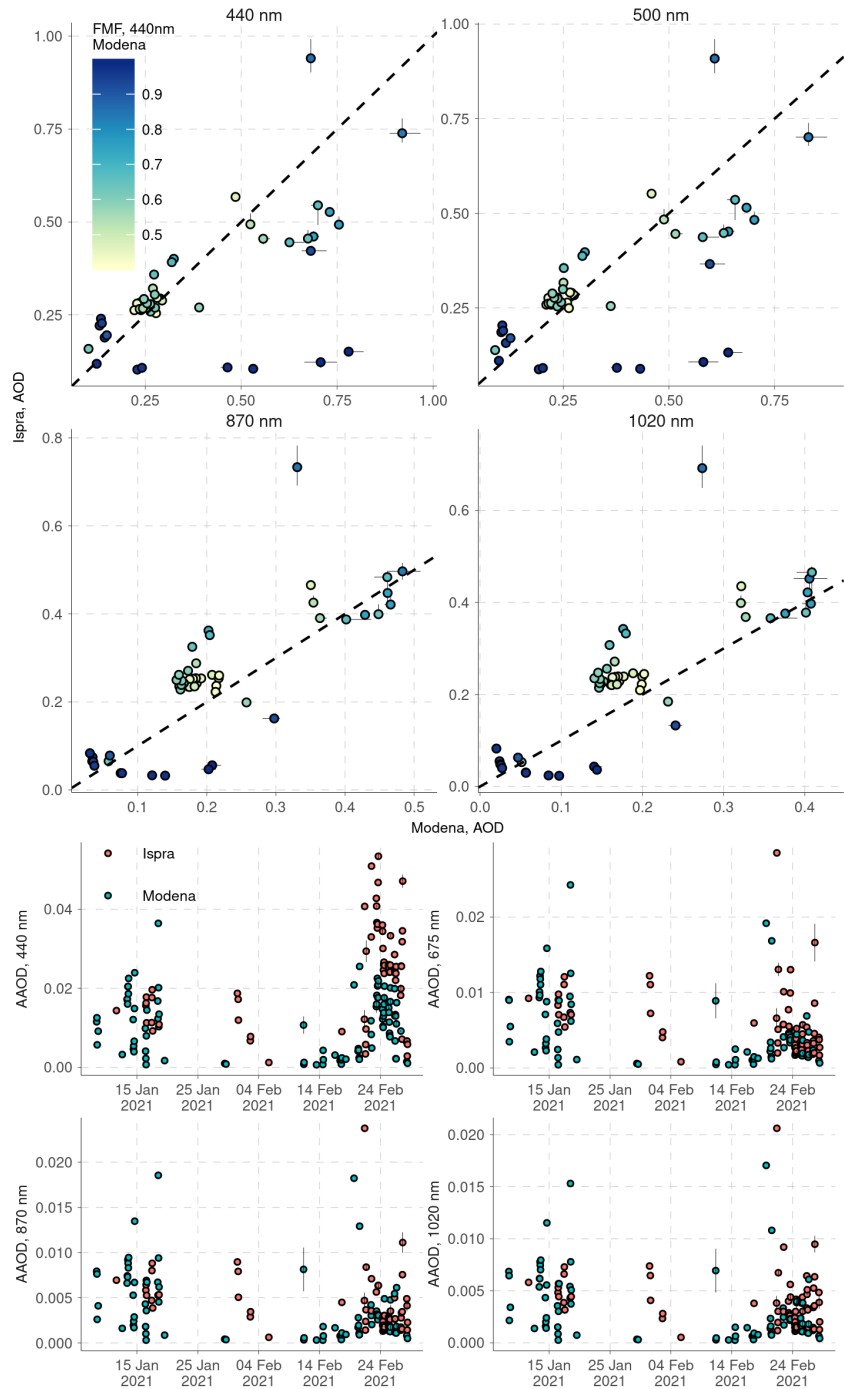

**Figure 10.** Top panel: comparison of hourly median AOD retrieved in Modena and Ispra at 4 wavelength during the investigated period (January 2020 – March 2021), color-coded according to the Fine Mode Fraction (FMF) at 440 nm. The dashed line indicates the 1:1 line. Lower panel: timeseries in Modena and Ispra of AOD and FMF at 440 nm during the investigated period (January 2020 – March 2021). The bars indicate the hourly interquartile range.

**Appendix B:  Symbol and acronyms**

**Table B1.** Description of symbols and acronyms used in the text.

| Symbol | Description |
|---|---|
| $\sigma_{ap}$ | Aerosol particle absorption coefficient |
| $\sigma_{ap}^{BC,\,ff}$ | $\sigma_{ap}$ from BC by fossil fuel combustion |
| $\sigma_{ap}^{BC,\,bb}$ | $\sigma_{ap}$ from BC by biomass burning |
| $\sigma_{ap}^{BrC}$ | $\sigma_{ap}$ by Brown Carbon |
| | |
| AAE$^c$ | Absorption Ångström exponent for column observations |
| AAE$^c$1 | AAE$^c$ across the 440 – 675 nm range |
| AAE$^c$2 | AAE$^c$ across the 675 – 880 nm range |
| AAE$^i$ | Absorption Ångström exponent for in-situ observations |
| AAE$^i_{ff,BC}$ | AAE$^i$ from BC by fossil fuel combustion |
| AAE$^i_{BrC}$ | AAE$^i$ from BrC |
| AAOD$_{\lambda,\,species}$ | Absorption AOD at wavelength $\lambda$ and *species* (i.e. BC, BrC or dust) |
| ADRE | Absorptive DRE |
| AERONET | AErosol RObotic NETwork |
| AOD$_\lambda$ | AOD at wavelength $\lambda$ |
| ApAOH | Apparent Aerosol Optical Height |
| BOA | Bottom of the atmosphere |
| BC | Black Carbon |
| BL | Boundary layer |
| BrC | Brown Carbon |
| DRE | Direct radiative effect |
| eBC | equivalent Black Carbon |
| FMF$_\lambda$ | Fine mode fraction at wavelength $\lambda$ |
| HR | Heating rate |
| IQR | Interquartile range |
| LAA | Light absorbing aerosol |
| ME | Mean error |
| MLH | Mixing layer height |
| MNE | Mean normalised error |
| MWAA | Multi-wavelength absorption analyzer |
| SAE$^c$ | Scattering Ångström Exponent for columnar observations |
| SAE$^c$1 | SAE$^c$ across the 440 – 675 nm range |
| SAE$^c$2 | SAE$^c$ across the 675 – 880 nm range |
| SAOD | Scattering AOD |
| SSA | column single-scattering albedo |
| TOA | Top of atmosphere |
| UB | Urban background |
| UT | Urban traffic |

*Code and data availability.* We provide:

– the implementation in R programming language of the dual spot correction algorithm following Drinovec et al. (2015), which was used for 1-minute urban background data (Bigi, 2023, version 1.0.0 at http://doi.org/xxxx/zenodo.xxxx, last access: xxx). This open-source code is distributed under the BSD-3 License.

– the R code for the apportionment of the AERONET data according to Bahadur et al. (2012) (Bigi, 2023, version 1.0.0 at http://doi.org/xxxx/zenodo.xxx, last access: xxx). This open-source code is distributed under the BSD-3 License.

– Raw *in-situ* absorption data for Modena (Bigi, 2023, version 1.0.0 at http://doi.org/xxxx/zenodo.xxx, last access: xxx). The latest version of these tools are available in dedicated GitHub repositories (https://github.com/abigmo/ae33_dualspot_correction and https://github.com/abigmo/aaod_apportionment, last access: XXX).

AERONET data are publicly available at the https://aeronet.gsfc.nasa.gov/. Regulatory air quality data for Modena are publicly available both at https://dati.arpae.it/ and on the European Environmental Agency air quality portal. Meteorological data for Modena are publicly
available at https://dati.arpae.it/.

*Author contributions.* AB designed the study, acquired the funds for the absorption in-situ measurements, led the writing of the manuscript and the data analysis. GV, EA, MCC, VB, DM, LF and GG contributed to the development of the methodology and to data interpretation. ST and LG funded and maintained the sun photometer. DM ran the MWAA code. LF computed the HR. All authors contributed to the manuscript.

*Competing interests.* The authors declare no competing interests.

*Acknowledgements.* This study was supported by the project 'Black Air' (CUP E94I19001080005) funded by the University of Modena and Reggio Emilia and the *Fondazione di Modena* under the programme "Fondo di Ateneo per la Ricerca 2019". These results were also obtained within the MUSA – Multilayered Urban Sustainability Action – project, funded by the European Union – NextGenerationEU, under the National Recovery and Resilience Plan (NRRP) Mission 4 Component 2 Investment Line 1.5: Strenghtening of research structures and creation
of R&D "innovation ecosystems", set up of "territorial leaders in R&D". LF acknowledges the GEMMA Center in the framework of the project TECLA, MIUR 'Dipartimenti di Eccellenza 2023-2027'. EA was supported by NOAA cooperative agreements NA17OAR4320101 and NA22OAR4320151. Carla Barbieri and Enrica Canossa from ARPAE are kindly acknowledged for hosting the filter absorption photometers in the air quality monitoring stations and for granting full access to these sites. The municipality of Modena is kindly acknowledged for providing the vehicular traffic data. Hersbach et al. (2018) was downloaded from the Copernicus Climate Change Service (C3S) Climate
Data Store: the results contain modified Copernicus Climate Change Service information 2020 and neither the European Commission nor ECMWF is responsible for any use that may be made of the Copernicus information or data it contains. We thank Giuseppe Zibordi and his staff for establishing and maintaining the Ispra AERONET site used in this investigation.

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
