# Peer review of "Aerosol absorption by in-situ filter-based photometer and ground-based sun-photometer in a Po valley urban atmosphere"

_EGUsphere, 2023_

## Referee Comment (RC1)

**General comment**

The manuscript by Bigi et al. analyses aerosol absorption data from both in situ and passive remote sensing collected in the Po Valley, complemented by ancillary datasets of both atmospheric composition and meteorological records. Although limited to few measuring sites and to a relatively short period, it presents new data that can be of interest for the scientific community. The manuscript is sufficiently well structured and addresses relevant scientific questions within the ACP scope.

There are, however, some major weaknesses I would like the authors to address before recommending the manuscript for publication.

These are listed below followed by some additional details and/or minor comments

**Major points:**

I) A major drawback of the current data presentation is that the reported measurements refer to different periods depending on the instruments and site. It is thus not always clear if the results derived by different instrument at different sites, and summarized in Figures, are actually directly comparable. For example: are the diurnal cycles of quantities presented in Figure 2 and 3 based on data collected in the same periods? Is this true at both sites? A Table summarizing clearly the datasets used in the analysis and presented in the different Figures would be helpful. This table should also include clear indication of data filtering, when applicable. In fact, it is not always clear comparing Figures which data were filtered and how.

For example, in Section 2.2 the authors state that *'Due to these limitations, in the present study, retrievals during events of high altitude dust transport were discarded from the analysis.'* Similarly, in Section 2.4.1 the authors state that '*the days with significant dust load were discarded prior the application of the MWAA model to the in-situ data. Days with significant dust content were first identified for the atmospheric column, using the particle volume size distribution estimated by the AERONET inversion (Sinyuk et al., 2020), and subsequently compared with HYSPLIT back trajectories. Additionally, the impact of dust at ground levels on these days 255 was evaluated using the daily PM2.5 PM10 ratio by the in-situ measurements (Figure S1).*' However, their Figures 6, 7 and 9 clearly show that dust affects most of the data-points presented there.

Thus a Table with clear indication of the datasets used in the analysis, and data filtering if any, would be beneficial.

II) A second aspect that would merit further explanation is the use of MLH data in this work. These come from ERA 5 (Sec 2.1) and are quite key in several parts of the analysis. As a first suggestion, I would encourage providing some more info on the ERA5 dataset used (e.g., spatial and temporal resolution,…). Then, I think it would be important to understand how this model-based information is representative for the sites under investigation (given that, if I am not mistaken, ERA5 spatial resolution is 25km). In Section 2.3, the authors report an expected error in the MLH of 50 m, but some additional comments could be added in relation to the expected error in the specific investigated site (for example some previous experimental studies measuring MLH and/or PBLH in the area could be used to this purpose). Additionally, in Section 3.2 the authors seem to attribute the discrepancies obtained (e.g. Figure 6) to an erroneous MLH estimate, so I think the authors should better comment on the use of this dataset.

III) The two weak points above combine into a particularly critical Section 3.2. To my opinion, the scientific methods and assumptions in this section are not fully valid and clearly outlined, as further detailed in the comments on Section 3 below (points 28-30). I would suggest considering eliminating this Section 3.2 of the manuscript, or, if not, revise and clarify it taking into account the relevant specific comments below.

**Specific comments and minor suggestions**

**Introduction**

1. First sentence introduces black carbon (BC) and states that this is often reported as 'equivalent black carbon' (eBC). Given the core subject of this paper, I think it would be important to explain here in which way BC and eBC are not actually synonyms and why the term 'equivalent' was therefore introduced.
2. Line 30: Better to use ' in the range' here.
3. Line 31: To be rephrased, removing 'resulted'
4. Line 83 Should better be: '..tropospheric layer. However, estimating the vertical distribution of aerosol or their columnar load..'. (In fact, strictly speaking AERONET does not measure the aerosol vertical distribution).
5. Line 52: Maybe the reference here could be updated with a more recent one
6. Lines 89-91: Although I understand what the authors mean, the sentence is not clear enough, please rephrase.
7. Line 92: Not sure 'by' is the correct preposition here.
8. Line 98: the study area is never introduced before in the text (it only appears in the Abstract), so this sentence should be rephrased or the study area should be introduced in the text before this statement.
9. Lines 111-113: Can you be more specific by quantifying the impact?

**Section 2**

10. I would suggest modifying slightly the title of the section (e.g. 'Measurement site and methods').
11. Line 174:  I think the MLH dataset should not be introduced and described within the 'in situ surface measurement' section, being this not derived by in situ measurements but from global modelling.
12. Line 180: refer to Figure 1 here
13. Lines 195-198: the use of the term 'atmospheric thickness' seems incorrect here. If I understand, this is rather the thickness of the aerosol-loaded layer.
14. Line 202: It is not clear to me where this 15% comes from.
15. Lines 205-210: Sentence not clear, please rephrase. It seems from the previous discussion that HR depends on the thickness of the investigated layer, while this sentence states the opposite.
16. Line 211: how did you evaluate periods of desert dust transport? If this was done as described in Section 2.4.1, please refer to that section here.
17. I suggest to remove Section 2.3 and move discussion of uncertainty to the two relevant sections.
18. Both Section 2.4.1 and 2.4.2 should include at the beginning a brief overview of the methods. Some e readers may be not familiar of the principles of AAE- and SAE- based sources apportionment. This is indeed what is expected in the 'methods section'. Although the readers can be usefully referred to relevant literature, these methods should be at least briefly explained in this section (for Section 2.4.2, text now in Appendix A could be used for the purpose).
19. Lines 251-255: can you provide more details? Using the particle size distribution how? using backtrajectories how? Which threshold on the PM2.5/PM10 ratio has been used (this is not clear from Fig. S1).
20. Lines 270-275: the choice of the datasets used to derive AAE for BC and AAE for BrC is critical, in particular use of two completely different seasons may bias somehow the results. Also, it is clear that statistical significance of the two datasets is quite different (1782 data points vs 89 data points), can you further comment on that?
21. Lines 284-289: This paragraph is unclear. Please, rephrase to explain better. Figure S3 also clearly highlights the intense dust episode occurred at the end of February. It is also not clear to me if the 'biased' values of AAE BC and AAEBrC that are clearly dust affected have been excluded from the statistics in Table 1 or not.

**Section 3**

22. Line 295: What about Saturday's data? Also note a typo error for 'panel' here.
23. Lines 302-323: This part is basically a comparison with available literature. I think this extended text could be better summarized in a Table comparing results of this study with relevant literature (as done for the AERONET-based statistics).
24. The role of residential heating (other than BB) should be also commented when discussing the diurnal cycles of Figure 2. It is rarely mentioned within the text.
25. If kept (see main comment III above), Section 3.2 title should be revised (e.g. Comparing absorption optical depth from remote sensing and in situ values).
26. Line 392: replace 'were' with 'was'
27. Lines 400: Note that label 'a)' in the list is missing.
28. Lines 400-405: To my opinion the results in Figure 6 clearly suggest one main point: for the species showing higher absorption at longer wavelengths (880, 675 nm) assumption of a uniform absorption distribution through MLH is not valid. In fact, reasons for discrepancies listed by the authors as b) and c) would also affect results in the blue range (third panel), which is not the case.
29. Also note that definition and use of ApAOD is critical. First, it has the dimensions of a length; therefore the naming of this quantity seems not particularly suitable. Additionally, its definition assumes that $AAOD = abs_{insitu} \times ApAOD$, which might be not the case for example in cases of transport of elevated aerosol layers, as during dust transport events, in which aerosol layering might be different (e.g. $AAOD = abs_{insitu} \times H_{layer1} + abs_{aloft} \times H_{Layer2}$ )
30. Lines 425-434: It is certainly true that '*Differences in AAOD might originate from a different atmospheric layering and mixing*' but this implies that statement at lines 425-433 needs to be rephrased. Although I understand this claim, in the current formulation the statement is not correct. In fact, the decrease in PM2.5 / PM10 ratio only indicates that some dust is reaching the ground, but is not telling anything about its vertical distribution.

**References**

31. References are often given in non-chronological order within the text. Please, check the journal instructions on this.
32. Section 2.2 should include reference to O'Neill et al. papers, on which most AERONET almucantars retrievals are based on.

**Figures:**

33. I think panels within Figures should be numbered (or labelled) and Figure captions should refer to this numbering (labelling) to be more readable. Please, check the journal instructions on this.
34. Figure S2: please improve color scale range, as in the current form it seems not optimal for the dataset presented (little variability among data dots color)

**Tables:**

35. Table 1: why reporting medians and associated median absolute deviance? Generally, when reporting median values, associated percentiles (typically 25-75$^{th}$) are rather included. Are you sure the same statistical parameters are reported in the referred literature? Given the wavelength dependence of the AAE, probably the table could be restructured separating values by spectral range, making the table more readable.
36. Table 2: specify wavelength rather than 'IR', for homogeneity with the text and Figures
37. Table 3: I would remove this one. As well documented in the text, most variables have a marked diurnal cycle, providing Median and interquartile values here could be misleading and hide important information on daily variability. I would rather suggest to insert Figure S4 in the main text, showing the diurnal cycles of the same quantities (as done for the atmospheric composition ones).
38. Table A1 now is given before introducing Appendix A, and is not related to it

**TITLE:** A further suggestion is to include reference to the study area in the manuscript title, as it seems a bit too general in its current form.

**Language:** The language is not always fluent and precise, and some revision by a mother tongue is encouraged.

---

## Author Comment (AC2)

We thank Reviewer 2 for his/her positive remarks, suggestions and hints, which allowed us to improve the overall quality of this study.

In the reply we used blue fonts for the reviewer's text, black fonts for the authors' reply and green italic fonts for the excerpts of the revised manuscript.

*The manuscript by Bigi et al. attempts to bridge between in situ measurements (obtained through with multi-wavelength absorption photometry) and AERONET sunphotometer retrievals, in the air pollution hotspot of the Po Valley. Established modelling approaches are followed for the disaggregation of BC-BrC components and also for the apportionment of BC fossil fuel and biomass burning sources at urban traffic and background locations. The insights gained by the paper are important in terms of how much in-situ absorption monitoring translates into actual absorption aloft and whether it can be used for radiative forcing estimates. The characterization of the intra-urban variation of absorbing aerosol especially between different site types can lead also to useful inferences in the ambient exposure domain, thus, the authors are encouraged to expand towards the analysis of intersite associations/contrasts for in-situ source-specific BC and BrC. There should be also some attempt to discuss the transport of absorbing pollutants within or from outside the Po Valley. The paper is generally clearly written, with pertinent references and adequate discussion. It also recognizes uncertainties and limitations of the monitoring/modelling approaches used. It can be considered for publication after exploring the above mentioned directions and addressing the following specific comments.*

**Specific Comments**

*Introduction, 3rd paragraph: The health effects from exposure to BrC should also be mentioned, given that a substantial part of light-absorbing OC are of polycyclic aromatic nature and therefore linked to oxidative stress induction and carcinogenic effects.*
We agree with the reviewer and in the Revised Manuscript (RM) we added the following statement (line 52): *BrC has also been shown to have detrimental health effects, enhanced because of its enrichment in organic compounds (Chowdhury et al., 2019; Offer et al., 2022), possibly related to aerosol aging (Li et al., 2022; Tuet et al., 2017; Weitekamp et al., 2020).*

l48-49: Given the existing ambiguity in the definition of eBC, this compilation should be recognized as a rather challenging task.
In the RM we changed these lines as follows (line 56): "*[...] the large uncertainty associated with source emission factors, PM speciation and eBC definition makes the implementation of systematic and harmonized emission estimates a challenging task.*"

l59-65: Why go into such details about these methods when they are not used in the present study?

While there are extra (and perhaps extraneous) sentences on the details about these methods, we consider it useful for the reader since there is quite a lot of active research on BC measurement techniques.

l101: Sources such as? Discuss.

In the RM the sentence was modified as follows (line 114): *Significant aerosol sources other than traffic remain present in the valley, e.g. biomass burning by domestic heating for several compounds including organic aerosols and BC, and farming for $NH_3$, a major PM precursor. Their role in PM levels was highlighted by the small decrease in PM across the basin (Ciarelli et al., 2021; Putaud et al., 2021) and in particle count in Modena (Shen et al., 2021) during the 2020 lockdown due to the SARS-CoV-2 pandemics.*

l132: Give some more details about the fuel types that are used for domestic heating and emit PM in the area. The impact of residential biomass burning should be introduced because BB aerosols are integral to this study.

In the RM more details regarding domestic heating emissions have been provided (line 153): *More specific to non-industrial combustion (SNAP 2), most of buildings use compressed natural gas for both heating and cooking; consistently 99.4% of $PM_{10}$ emissions by SNAP 2 are estimated to be produced by biomass combustion for domestic heating (ARPAE, 2020).*

l134-136: This statement might be a bit ambitious, and taken out of context depending on how the reader understands the extent of the Po Valley. Moreover, the limited impact of industry that the authors claim for the Modena area, might not be the case in other more industrialized cities. I would suggest to tone this part down.

In the RM the sentence was changed according to the following (line 158): *Modena's setting is quite representative of several mid-size urban areas across the Po valley, particularly in terms of traffic and domestic emissions sources and topography.*

l142-143: Indicate the distance from the nearby road for the UB site, and the traffic intensity in the adjacent road for the UT site. It could be also useful to mention some inter-site differences observed for the regulatory pollutants, to illustrate how clear the UT-UB distinction is.

This information are now included in the RM (line 166): *The UT site faces a major road with two lanes per direction, with estimated median daily traffic counts of ~ 20 thousand vehicles, while the UB is within Modena's largest urban park at a distance of ~120 metres from the nearest road. [...] The daily $PM_{10}$ median (10th, 90th quantiles) concentration at the UB site over the period 2017 – 2021 was 24 $\mu g\ m^{-3}$ (13 $\mu g\ m^{-3}$ , 57 $\mu g\ m^{-3}$), while at the UT sites the same statistics for $PM_{10}$ resulted were 27 $\mu g\ m^{-3}$ (14 $\mu g\ m^{-3}$ , 63 $\mu g\ m^{-3}$). Consistently, over the same period, hourly $NO_2$ at the UB showed lower levels than at the UT site, with the two locations having a median (10th, 90th quantiles) of 23 $\mu g\ m^{-3}$ (6 $\mu g\ m^{-3}$ , 50 $\mu g\ m^{-3}$) and 35 $\mu g\ m^{-3}$ (14 $\mu g\ m^{-3}$ , 66 $\mu g\ m^{-3}$) respectively.*

l155: The aggregation procedure here is unclear. What is meant by "custom"?

We agree wth the reviewer that 'custom' is not a proper definition. The aggregation of the raw transmittance count from the MA200 is based on a transcription in R programming language of the dual-spot compensation algorithm presented by Drinovec et al. (2015). In the RM the text was changed as follows (line 179).

*In order to compensate for the occasionally low absorption readings at the latter site, the 1-minute raw transmittance counts at UB were firstly aggregated to 5 minutes and then used to compute the corresponding $\sigma_{ap}$ by a transcription in R programming language of the dual-spot compensation algorithm as described in Drinovec et al. (2015)*

l157: A flow rate over 100cc is necessary to use the DualSpot compensation, but what factored in using different seasonal flow rates? Is there some recommendation by the authors?

The larger flow in summer is needed because of the lower atmospheric levels: if one wants to keep the same time resolution, flow needs to be increased to deposit sufficient material on the filter. In the RM the sentence was changed according to the following (line 182): *Flow was set to 100 ml min$^{-1}$ in winter and increased to 125 ml min$^{-1}$ in summer, because of the lower atmospheric concentrations.*

l210-211: Describe in brief the screening process.

We screened them referring to work done in section 2.4.1. Thus, as suggested also by reviewer 1, in the RM we now refer to this section regarding this screening issue (line 281).

Section 2.3: A separate uncertainty section is not necessary here. Mover this information to the respective sections.

In the RM the uncertainty of each dataset (in-situ observations, ERA5 model and columnar observations) is now included in the respective method section.

Section 2.4.1: I understand that the model has been presented already in literature, however, it is necessary to include a description of the procedure here and maybe some of the key equations. Otherwise, the reader will not understand how the different AAE values that are preselected                    are                put                into                use.

In the RM the description of the in-situ and of the columnar absorption data is now expanded, as                              reported                              below.

line 295: *In-situ aerosol absorption coefficient $\sigma_{ap}(\lambda)$ was apportioned to species (i.e. Black Carbon, Brown Carbon, referred to as $\sigma_{ap,BC}(\lambda)$ and $\sigma_{ap,BrC}(\lambda)$, respectively) and sources (fossil fuel and biomass burning combustion, referred to as $\sigma_{ap,FF}(\lambda)$ and $\sigma_{ap,BB}(\lambda)$, respectively) using the Multi-Wavelength Absorption Analyzer model (MWAA model, Massabò et al., 2015; Bernardoni et al., 2017). This model assumes an equivalence between the Absorption Ångström Exponent (AAE, Moosmüller et al., 2009) of BC and that of fossil fuel ($AAE^i_{FF}=AAE^i_{BC}$), and it assumes biomass burning to be the only source of BrC. Under these hypotheses, the MWAA model assumes that both the following equations hold for the total $\sigma_{ap}(\lambda)$ at each wavelength:*

$$\sigma_{ap}(\lambda) = \sigma_{ap} BC(\lambda) + \sigma_{ap} BrC(\lambda) = A\lambda^{-AAEiBC} + B\lambda^{-AAEiBrC} \qquad (2)$$
$$\sigma_{ap}(\lambda) = \sigma_{ap} FF(\lambda) + \sigma_{ap} WB(\lambda) = A'\lambda^{-AAEiFF} + B'\lambda^{-AAEiBB} \qquad (3)$$

*In Equations 2 and 3 $AAE^i_{BC}=AAE^i_{FF}=1$ was set, based on the $AAE^i$ computed over 5 wavelengths at morning rush hour on winter weekdays at UT, consistent with fresh uncoated BC particles (e.g. Liu et al., 2018). $AAE^i$ for BrC was determined by a preliminary non-linear fit of Equation 2, performed considering $AAE^i_{BrC}$ as a free parameter (and resulting in an average $AAE^i_{BrC}=3.9$); $AAE^i_{bb}=2$ was set on literature data for the Po valley (Bernardoni et al., 2011, 2013; Vecchi et al., 2018; Costabile et al., 2017). A, B were then obtained for each sample by multi-wavelength fit of Equation 2 (after fixing $AAE^i_{BrC}$) and A', B' by multi-wavelength fit of Equation 3. It is noteworthy that MWAA hypotheses neglect possible contributions from mineral dust. To limit uncertainties resulting from this, the days with significant dust load were discarded prior the application of the MWAA model to the in-situ data, i.e. whenever the in-situ apportionment data is presented throughout the text, it is screened for dust.*

line 321: *AAOD was apportioned to BC, BrC and mineral dust using the approach proposed in Bahadur et al. (2012), i.e. by directly solving the system of Ångström equations (see Appendix A) using the AERONET almucantar L1.5\* retrievals. The system includes Equations A1, reporting the additive contribution of AAOD by each species to the total AAOD and Equations A2, reporting the exponential dependence of AAOD on the wavelength.*

l245: Do you consider that there are uncertainties around this assumption? For example, at a traffic impacted location, there could be some traffic-related BrC expected. This would also probably mean that AAE-BC and AAE-FF are not identical. Please discuss and recognize the limitations.

The model used to process data presented in this work is based on the hypothesis that $AAE_{BC}=AAE_{FF}=1$. The Referee is certainly raising an interesting and complex point, since it is possible that these two parameters have different values depending, for example, on the aerosol aging or even on the type of fossil fuel burned. However, the effect of their variation on both optical and source apportionments is of second order to the more critical parameters such as $AAE_{BrC}$ and $AAE_{BB}$. The limitations related to the choice of these parameters ($AAE_{BC}$; $AAE_{FF}$) are the same connected to source apportionment methods based on optical properties. We strongly think that a sensitivity study able to evaluate the effects of the variation (and linked uncertainties) of these parameters is beyond the scope of the present work; a dedicated study is currently underway and will soon be proposed as an independent paper. It will also present and discuss a freeware version of a software that integrates an updated and extended version of the MWAA model.

l250: The selection of a fixed AAE-BrC value is critical for the calculations and it has to be justified better here. Provide more information on how the 3.9 value was derived (method, location, season, dominant sources etc.).

As now better explained at line 304 of the RM, 3.9 was the average value obtained on the dataset presented in this paper, running the MWAA model having AAE-BrC as a free parameter.

l292-301: I don't think that a whole paragraph introducing the Figures is necessary. You can

guide the reader through the presentation of the results.

We think that  the dataset can be slightly tricky so a short recap paragraph is useful to present the analysis.

l302-303: Mean values are mentioned here, while in the Figure-Table, the medians are displayed. Maybe consider a homogenization of the presentation.

We agree with the reviewer, but since we compared the data to existing literature, depending on the referred article we had to compute the corresponding statistics on our dataset, in order to be comparable.

Figure 2: A couple of things stand out here and should be discussed. First, in the holidays the nighttime peak of BCff absorption at UB is comparable to the workdays, and also bigger than that at UT. Second, at the UT site, nighttime BCbb and BrC absorptions become larger in holidays than in workdays.

We agree with the reviewer that these are two noteworthy points and have been added in the RM at line 395: *More specifically, on weekday evenings $\sigma_{ap,BC,ff}$ peaks at 20:00 LT, one hour later than on holidays, at both UB and UT, with the former site recording $\sigma_{ap,BC,ff}$ levels higher in the evening that in the morning."* and lines 408: *"The weekly pattern for these two species is larger at the UT, with an increase during holidays in the overall median values of $\sigma_{ap,BC,bb}$ and $\sigma_{ap,BrC}$ of 22% and 35% respectively, along with an increase in their IQR of 16% and 28%. [...] The holiday increase in biomass burning aerosol is probably linked to the longer stay at home compared to weekdays and to a large recreational use of biomass burning in town*

l302-304: Is there an increasing interannual trend for absorption in cities of the Po valley? Discuss.

In the RM a note regarding Elemental Carbon trend is now included within the Introduction section (line 112). *Similarly, a drop of ~4% per year over the period 1997 – 2016 was recorded for the elemental carbon content in fog samples at the rural background site of San Pietro Capofiume (Gilardoni et al., 2020b).*

l321: It should be "BrC estimated".

In the RM that paragraph was rewritten and most of the text was moved to a table.

Table 2: The vehicle count parameter should be expressed in vehicles-per-day units.

We agree with the reviewer and in the RM Table 2 reports the statistics regarding the total number of vehicles per day.

l350-355: Not much new in this paragraph and not in the core of the study. I suggest omitting it or condensing it to a sentence.

As correctly suggested by the reviewer, we condensed this paragraph into the following sentence (line 419) *$O_3$ exhibits a 'weekend effect' (Cleveland et al., 1974), common to most urban areas in Europe having a VOC–limited regime, i.e. on holidays ozone rises earlier in the morning due to the lower $NO_x$ levels, leading to a more efficient photocatalytic cycle, and drops later in the evening due to the (later) increase in $NO_x$.*

l380-387: The enhancement of BrC for winds of the southern direction should be explained, since in this study BrC is considered as a source-specific variable (BB-related). It can be observed that the winds related to the increase are only moderate. So it should be examined if there is indeed a BB source area or it is a low-wind stagnation effect during nighttime when the highest BrC levels are expected (it should be also noted that it is observed only in holidays).

It is very challenging to detect precisely the location of the source causing the increase in BrC at the UT site on holidays. The increase occurs on Sundays or holidays, consistently with the hypothesis of biomass burning from domestic heating for recreational use, and it occurred only in the evening/night and not during daytime: so the combination of domestic heating and the nighttime stagnation probably are both responsible for the increase in BrC. In the RM a short note about this was added (line 453): *Also at the UT $\sigma_{ap,BrC}$ is higher during holidays and under southerly winds. This latter increase occurs during evening/night hours (not shown), consistently with biomass burning from domestic heating for recreational use, with the increase probably enhanced by nighttime atmospheric stagnation.*

l396-398: Did you consider compensating for the wavelength discrepancy by adjusting in situ absorptions by the calculated absorption AAE?

We preferred not to compensate for relatively small wavelength differences, since the uncertainty in the AAE might introduce a larger error than the error proceeding from the direct comparison of two slightly different wavelengths.

Conclusions: The section repeats numerical results from the previous parts of the manuscript. Some more implications of the findings, regarding atmospheric absorption research and urban BC exposure should be added.

In the RM a further point regarding implications and outlooks was added (line 587): *An improved knowledge of the role by the in-urban emissions of LAA is critical to control local air quality, urban heat island effects and climate forcing and an apportionment of LAA based on their atmospheric levels, as presented here, contributes towards this goal. This study provides important insights on the role of the in-situ absorption monitoring in estimating the actual absorption aloft and whether it can be used for radiative forcing estimates. Moreover the characterization of the intra-urban variation of absorbing aerosol based on different site types contributes in the ambient exposure domain. Towards this latter outcome, a more in depth investigation of the contribution of urban areas to atmospheric LAA can be gained by the application of specific atmospheric dispersion tools, and this represents one of the major study outlooks. More specifically, Lagrangian particle dispersion models would provide information on atmospheric levels across the urban area at a fine spatial resolution, supporting advanced exposure studies, and, further, would give an estimate of the spatial- and time-resolved emission factors for LAA in the urban area.*

Technical corrections

l31: Delete "over the Earth"
In the RM the text was changed accordingly.

l33: Delete "depending…specifics"
In the RM the text was changed accordingly.

l43" That should be "increased eBC concentrations". Same at next line
In the RM the text was changed accordingly.

l314: "good agreement"
In the RM the text was changed accordingly.

l400: "overestimation"
In the RM the text was changed accordingly.

l400: "by some concurrent conditions: (a)"
In the RM the text was changed accordingly.

l402: "at the rural…"
In the RM the text was changed accordingly.

l405: "contribution"
In the RM the text was changed accordingly.

l450: "mean absolute deviation"
We actually meant the "median absolute deviation", the median of the absolute deviations from the data's median. We used this as a robust estimate of the variable dispersion, i.e. a robust

l479: "with Ispra and Modena"
In the RM the text was changed accordingly.

---

## Author Comment (AC3)

We thank Reviewer 1 for his/her thorough reading of the paper and for all the indications and suggestions provided. These highlighted unclear sections and weaknesses in the original manuscript which we believe to have solved in the revised manuscript. We sincerely thank Reviewer 1 for these hints, which allowed us to improve the overall quality of this study.

In the reply we used blue fonts for the reviewer's text, black fonts for the authors' reply and green italic fonts for the excerpts of the revised manuscript.

*The manuscript by Bigi et al. analyses aerosol absorption data from both in situ and passive remote sensing collected in the Po Valley, complemented by ancillary datasets of both atmospheric composition and meteorological records. Although limited to few measuring sites and to a relatively short period, it presents new data that can be of interest for the scientific community. The manuscript is sufficiently well structured and addresses relevant scientific questions within the ACP scope. There are, however, some major weaknesses I would like the authors to address before recommending the manuscript for publication. These are listed below followed by some additional details and/or minor comments*

*Major points:*

*- I) A major drawback of the current data presentation is that the reported measurements refer to different periods depending on the instruments and site. It is thus not always clear if the results derived by different instrument at different sites, and summarized in Figures, are actually directly comparable. For example: are the diurnal cycles of quantities presented in Figure 2 and 3 based on data collected in the same periods? Is this true at both sites? A Table summarizing clearly the datasets used in the analysis and presented in the different Figures would be helpful. This table should also include clear indication of data filtering, when applicable. In fact, it is not always clear comparing Figures which data were filtered and how. For example, in Section 2.2 the authors state that 'Due to these limitations, in the present study, retrievals during events of high altitude dust transport were discarded from the analysis.' Similarly, in Section 2.4.1 the authors state that 'the days with significant dust load were discarded prior the application of the MWAA model to the in-situ data. Days with significant dust content were first identified for the atmospheric column, using the particle volume size distribution estimated by the AERONET inversion (Sinyuk et al., 2020), and subsequently compared with HYSPLIT back trajectories. Additionally, the impact of dust at ground levels on these days 255 was evaluated using the daily PM2.5 PM10 ratio by the in-situ measurements (Figure S1).' However, their Figures 6, 7 and 9 clearly show that dust affects most of the data-points presented there. Thus a Table with clear indication of the datasets used in the analysis, and data filtering if any, would be beneficial.*

The dataset could seem intricate at a first glance. In the revised manuscript (RM) we made it easier to track which dataset is presented. As stated in the original manuscript, since the apportionment to dust is not possible for the in-situ observation, whenever dust load was significant the in-situ apportionment was not performed: this means that, whenever the apportionment of the in-situ data is presented (in the RM Figures 2, 3, 6, 9 and Table 2), that data is filtered for dust. In the RM Figures 7, 8 the in-situ data is not apportioned, i.e. the data shown is not screened for dust. This point was made more clear throughout the revised manuscript, trying to balance clarity and text simplicity; note that it includes also an amendment addressing point #22.

Line 367 RM "*Figure 2 shows the medians and interquartile ranges of atmospheric species obtained from in-situ observations along with hourly traffic count from the induction loops closest to each monitoring site for winter (December, January and February) from early 2020 until March 2021. This*

*data is screened for days with non-negligible dust load, as specified in 2.4.1. The σap at 528 nm for winter weekdays (Monday through Friday) and winter holidays (i.e. Sundays, local and national holidays) is in the top 365 panel of Figure 2 and represents the absorption by aerosol at about 4 m above the ground. Saturdays are excluded due to their mixed signal between a holiday and a weekday."*

Line 440 RM: "*A conditional bivariate polar function was applied to NOx, σBC,ffap at 880 nm and σBrCap at 375 nm at both the UT and UB sites, to identify the position of potential emission sources (Figure 6) on weekdays and holidays, excluding days with significant dust load at ground.*"

*II) A second aspect that would merit further explanation is the use of MLH data in this work. These come from ERA 5 (Sec 2.1) and are quite key in several parts of the analysis. As a first suggestion, I would encourage providing some more info on the ERA5 dataset used (e.g., spatial and temporal resolution,...). Then, I think it would be important to understand how this model-based information is representative for the sites under investigation (given that, if I am not mistaken, ERA5 spatial resolution is 25km). In Section 2.3, the authors report an expected error in the MLH of 50 m, but some additional comments could be added in relation to the expected error in the specific investigated site (for example some previous experimental studies measuring MLH and/or PBLH in the area could be used to this purpose). Additionally, in Section 3.2 the authors seem to attribute the discrepancies obtained (e.g. Figure 6) to an erroneous MLH estimate, so I think the authors should better comment on the use of this dataset.*

An extensive section 2.2 "Mixing Layer Height" was added to the RM and it includes a summary description of ERA5, the details regarding its estimate of the MLH along with a description of the closest experimental soundings and a discussion about the uncertainty in MLH by weather prediction models. Modena is in topographically flat region and we extracted the ERA5 estimate at the closest grid point over a flat topography, 14 km north of Modena. The grid point is representative for a cell of 0.25° x 0.25° (corresponding to about 20 km x 28 km in the study area). The closest experimental soundings are available at San Pietro Capofiume, a rural background site 53 km East of Modena.

*III) The two weak points above combine into a particularly critical Section 3.2. To my opinion, the scientific methods and assumptions in this section are not fully valid and clearly outlined, as further detailed in the comments on Section 3 below (points 28-30). I would suggest considering eliminating this Section 3.2 of the manuscript, or, if not, revise and clarify it taking into account the relevant specific comments below.*

We believe that points I and II and all points below have been addressed and therefore we kept Section 3.2 in the RM. In the RM, MLH in Modena was derived by ERA5 and its uncertainty is discussed extensively (section 2.2); note that, more specifically, MLH was estimated also by experimental radiosounding, although only at 12 UTC and at 53 km distant from Modena. We believe that ApAOD was misunderstood by the reviewer: in the study we leverage ApAOD using the assumption of a vertically homogeneous aerosol layer to immediately test whether/which LAA were contained and well mixed over the MLH.

**Specific comments and minor suggestions**

*Introduction*

1. *First sentence introduces black carbon (BC) and states that this is often reported as 'equivalent black carbon' (eBC). Given the core subject of this paper, I think it would be important to explain here in which way BC and eBC are not actually synonyms and why the*

*term 'equivalent' was therefore introduced.*

In the RM, the first paragraph now includes an explanation for the term equivalent BC as follows (line 25): *A wide range of experimental techniques are available for the experimental measure 25 ment of BC, relying on different properties of LAA. In order to harmonize the terminology used to report the concentration of this species, the scientific community recommends reporting BC observations based on light absorption as equivalent BC (eBC, Petzold et al., 2013). EBC aerosol particles have fairly constant refractive index across the ultraviolet - infrared (UV – IR) range (Moosmüller et al., 2009). The eBC concentrations are converted into light-absorbing carbon mass concentration using the mass specific absorption cross section (MAC). Another type of LAA is Brown Carbon (BrC, Andreae and Gelencsér, 2006; Laskin 30 et al., 2015) which is the fraction of light-absorbing organic aerosol whose optical properties differ from those of BC, because of their enhancement in absorption towards UV wavelengths.*

2. *Line 30: Better to use ' in the range' here.*

The RM was changed accordingly to this suggestion.

3. *Line 31: To be rephrased, removing 'resulted'*

In the RM the text was rephrase to (line 34): *In terms of global impact, BC was shown to have a positive direct radiative effect at the Top-Of-Atmosphere (TOA) in the range of 0.71 – 0.82 $Wm^{-2}$ (Chung et al., 2012; Bond et al., 2013; Lin et al., 2014). Estimates of global direct effect were lower for BrC than for BC, in the range of 0.04 – 0.57 $Wm^{-2}$ (Feng et al., 2013; Lin et al., 2014; Saleh et al., 2014; Jo et al., 2016; Brown et al., 2018; Zhang et al., 2020). BrC concentrations are very spatially variable and concentrations depend on the study specifics.*

4. *Line 83 Should better be: '..tropospheric layer. However, estimating the vertical distribution of aerosol or their columnar load..'. (In fact, strictly speaking AERONET does not measure the aerosol vertical distribution).*

The RM was revised according to this suggestion.

5. *Line 52: Maybe the reference here could be updated with a more recent one*

The RM was revised accordingly to this suggestion and a citation of the study by Evangeliou et al., (2021) was added (line 59): *[...] followed by biomass burning and industry (European Environment Agency, 2013), as more recently confirmed by the analysis of the eBC emission change in Europe due to COVID-19 lockdowns (Evangeliou et al., 2021). Similar to BC …*

6. *Lines 89-91: Although I understand what the authors mean, the sentence is not clear enough, please rephrase.*

The phrasing was totally aligned with all publications cited in the current study; nonetheless,

for the sake of clarity, the text was changed to (line 97): *Several authors used the ratio between the surface in-situ aerosol mass concentration or aerosol absorption and the boundary-layer-height (i.e. they rescaled surface data over this atmospheric layer), and showed how this ratio underestimated sun-photometry observations of AOD or absorption AOD (AAOD) respectively (e.g. Bergin et al., 2000; Slater and Dibb, 2004; Aryal et al., 2014; Chauvigné et al., 2016; Chen et al., 2019).*

7. *Line 92: Not sure 'by' is the correct preposition here.*

   In the RM the sentence was modified as follows (line 102): […] *the representativity of surface in-situ measurements* […].

8. *Line 98: the study area is never introduced before in the text (it only appears in the Abstract), so this sentence should be rephrased or the study area should be introduced in the text before this statement.*

   In the RM a short introductory sentence was added in line 106 "*[…] in the northern hemisphere, particularly in the US and Europe, the aerosol absorption coefficient (σap) decreased over the last decade(s). More specific to the region of interest for our study, the Po valley, is a European hot-spot for atmospheric pollution situated in northern Italy. A previous work on the Po basin observed […]*"

9. *Lines 111-113: Can you be more specific by quantifying the impact?*

   A quantitative estimate is now added in lines 128: "*A Europe-wide assessment of urban air quality by Thunis et al. (2017), based on a simplified dispersion model, estimated a 57% contribution by in-city emissions to urban PM$_{2.5}$ in Milan, making this city the one with the largest self-contribution to local PM$_{2.5}$ across the European Union.*"

**Section 2**

10. *I would suggest modifying slightly the title of the section (e.g. 'Measurement site and methods').*
    We agree with the suggestion and the title of the section was changed accordingly

11. *Line 174: I think the MLH dataset should not be introduced and described within the 'in situ surface measurement' section, being this not derived by in situ measurements but from global modelling.*
    In the RM MLH is now described in the dedicated section 2.2

12. *Line 180: refer to Figure 1 here*

    The RM was changed accordingly to this suggestion.

13. *Lines 195-198: the use of the term 'atmospheric thickness' seems incorrect here. If I understand, this is rather the thickness of the aerosol-loaded layer.*

Thank you very much for your question. As reported in Ferrero et al. (2014) the ADRE is computed normalizing $\Delta DRE_{ATM}$ by the relevant atmospheric thickness $\Delta z$: $ADRE = \Delta DRE_{ATM}/\Delta z$

The ADRE represents the radiative power absorbed by the aerosol per unit volume of the atmosphere ($W\ m^{-3}$). Moreover, the ADRE can be directly related to the atmospheric heating rate (*HR*) as reported in eq. 1 at line 204 of the submitted manuscript.

From your question we understand that the misunderstanding comes from the fact that the thickness $\Delta z$ that hosts most of the LAA in the Po valley is the one described by the mixing layer height as stated at lines 198-201 and thus the one with the highest optical signal (see answer to question 14). Thus we rephrased the sentence as follows (line 264): *However, as demonstrated in Ferrero et al. (2014), a more useful parameter is the Absorptive DRE (ADRE) of atmospheric aerosol, which can be computed simply by normalizing $\Delta DRE_{atm}$ by the atmospheric thickness $\Delta z$ of the aerosol loaded layer hosting most of the LAA which is (in the Po Valley) the mixing layer height (MLH).*

14. *Line 202: It is not clear to me where this 15% comes from.*

Thank you for this question which is related to the previous sentence which was missing a part, i.e. the average amount of AOD in the free troposphere (5% from Ferrero et al., 2019). Thus in lines 267 of the RM, the text was changed as follows: *The advantage of using ADRE in the Po Valley environment in wintertime is that, in this case, most of the AOD signal is built up within the mixing layer, as shown by both Ferrero et al. (2019), who found that in Milan up to 87% of AOD signal was generated within mixing layer, 8% in the residual layer and 5% in the free troposphere as also reported by Barnaba et al. (2010), who found consistent figures at the Ispra background site. This means that if the thickness $\Delta z$ is the MLH, the ADRE will refer to that layer with an expected maximum overestimation of approximately 13% (i.e. roughly the amount of aerosol optical depth above the MLH).*

15. *Lines 205-210: Sentence not clear, please rephrase. It seems from the previous discussion that HR depends on the thickness of the investigated layer, while this sentence states the opposite.*

The HR depends on the atmospheric thickness in which the aerosol is loaded. This was a typo and in order to avoid misinterpretation, in the RM the sentence we rephrased as follows (line 278): *This approach is limited because HR can be obtained directly by the AERONET retrievals only if most of the AOD signal is built up within the mixing layer.*

16. *Line 211: how did you evaluate periods of desert dust transport? If this was done as described in Section 2.4.1, please refer to that section here.*

As you stated we referred to work done in section 2.4.1. Following your suggestion we referred to this section at line 211.

17. *I suggest to remove Section 2.3 and move discussion of uncertainty to the two relevant sections.*

In the RM the uncertainty of each dataset (in-situ observations, ERA5 model and columnar

observations) is now included in the respective method section.

18. *Both Section 2.4.1 and 2.4.2 should include at the beginning a brief overview of the methods. Some e readers may be not familiar of the principles of AAE- and SAE- based sources apportionment. This is indeed what is expected in the 'methods section'. Although the readers can be usefully referred to relevant literature, these methods should be at least briefly explained in this section (for Section 2.4.2, text now in Appendix A could be used for the purpose).*

In the RM the description of the in-situ and of the columnar absorption data is now expanded, as reported below.

line 295 *"In-situ aerosol absorption coefficient $\sigma_{ap}(\lambda)$ was apportioned to species (i.e. Black Carbon, Brown Carbon, referred to as $\sigma_{ap,BC}(\lambda)$ and $\sigma_{ap,BrC}(\lambda)$, respectively) and sources (fossil fuel and biomass burning combustion, referred to as $\sigma_{ap,FF}(\lambda)$ and $\sigma_{ap,BB}(\lambda)$, respectively) using the Multi-Wavelength Absorption Analyzer model (MWAA model, Massabò et al., 2015; Bernardoni et al., 2017). This model assumes an equivalence between the Absorption Ångström Exponent (AAE, Moosmüller et al., 2009) of BC and that of fossil fuel ($AAE^i_{FF}=AAE^i_{BC}$), and it assumes biomass burning to be the only source of BrC. Under these hypotheses, the MWAA model assumes that both the following equations hold for the total $\sigma_{ap}(\lambda)$ at each wavelength:*

$$\sigma_{ap}(\lambda)= \sigma_{ap}\, BC(\lambda)+ \sigma_{ap}\, BrC(\lambda)= A\lambda^{-AAEiBC}+ B\lambda^{-AAEiBrC} \qquad (2)$$
$$\sigma_{ap}\,(\lambda)= \sigma_{ap}\, FF(\lambda)+ \sigma_{ap}\, WB(\lambda)= A'\lambda^{-AAEiFF}+ B'\lambda^{-AAEiBB} \qquad (3)$$

*In Equations 2 and 3 $AAE^i_{BC}=AAE^i_{FF}=1$ was set, based on the $AAE^i$ computed over 5 wavelengths at morning rush hour on winter weekdays at UT, consistent with fresh uncoated BC particles (e.g. Liu et al., 2018). $AAE^i$ for BrC was determined by a preliminary non-linear fit of Equation 2, performed considering $AAE^i_{BrC}$ as a free parameter (and resulting in an average $AAE^i_{BrC}=3.9$); $AAE^i_{bb}=2$ was set on literature data for the Po valley (Bernardoni et al., 2011, 2013; Vecchi et al., 2018; Costabile et al., 2017). A, B were then obtained for each sample by multi-wavelength fit of Equation 2 (after fixing $AAE^i_{BrC}$) and A', B' by multi-wavelength fit of Equation 3. It is noteworthy that MWAA hypotheses neglect possible contributions from mineral dust. To limit uncertainties resulting from this, the days with significant dust load were discarded prior the application of the MWAA model to the in-situ data, i.e. whenever the in-situ apportionment data is presented throughout the text, it is screened for dust.*

line 321 *"AAOD was apportioned to BC, BrC and mineral dust using the approach proposed in Bahadur et al. (2012), i.e. by directly solving the system of Ångström equations (see Appendix A) using the AERONET almucantar L1.5* retrievals. The system includes Equations A1, reporting the additive contribution of AAOD by each species to the total AAOD and Equations A2, reporting the exponential dependence of AAOD on the wavelength."*

19. *Lines 251-255: can you provide more details? Using the particle size distribution how? using backtrajectories how? Which threshold on the PM2.5/PM10 ratio has been used (this is not clear from Fig. S1).*

In the RM more details were added to this point. Line 311: *Days with significant dust content were first identified for the atmospheric column, using the particle volume size distribution estimated by the AERONET inversion (Sinyuk et al., 2020); the identification of dust events was performed qualitatively, based on the retrievals having a dominant coarse mode (e.g.*

*Figure S7, panel b). These retrievals were subsequently double-checked by 72-hours HYSPLIT back trajectories using Global Data Assimilation System (GDAS) 1° resolution wind fields. Additionally, the impact of dust at ground level was assessed based on the daily $PM_{2.5}$ to $PM_{10}$ ratio from the in-situ measurements (Figure S1), with ratio $\leq 0.5$ as a qualitative threshold for a dust event. For reference, the daily $PM_{2.5}$ to $PM_{10}$ ratio in winter between 2017 to 2021 at the UB site had a median of 0.71 and a 10th (25th) quantile of 0.53 (0.62), i.e. the two aerosol fractions are quite similar, as already observed at most UB sites across the basin (Bigi and Ghermandi, 2016).*

20. *Lines 270-275: the choice of the datasets used to derive AAE for BC and AAE for BrC is critical, in particular use of two completely different seasons may bias somehow the results. Also, it is clear that statistical significance of the two datasets is quite different (1782 data points vs 89 data points), can you further comment on that?*

Thanks to the reviewer for this comment. In order to identify AAE for each species two different datasets were used. To select AAE of BrC only winter retrievals were used, which is when the maximum contribution from biomass burning occurs. Biomass burning is expected to be the main source of BrC, although in winter there are fewer AERONET retrievals, because of the shorter daytime, the presence of clouds etc. This is a further reason to use the quantile and the median absolute deviation. We agree that the assessment of the AAE for BC is more reliable, since it is based on a longer dataset, and it is more representative of freshly emitted BC.

21. *Lines 284-289: This paragraph is unclear. Please, rephrase to explain better. Figure S3 also clearly highlights the intense dust episode occurred at the end of February. It is also not clear to me if the 'biased' values of AAE BC and AAEBrC that are clearly dust affected have been excluded from the statistics in Table 1 or not. First sentence introduces black carbon (BC) and states that this is often reported as 'equivalent black carbon' (eBC). Given the core subject of this paper, I think it would be important to explain here in which way BC and eBC are not actually synonyms and why the term 'equivalent' was therefore introduced.*

In the RM the paragraph on the solution of equations A1, A2 is extended and improved as follows (line 353): *Finally, with reasonable confidence in the tailored AAE values for the different absorbing components, each AERONET retrieval at Modena and Ispra was apportioned by summarizing the solutions of the equation system as described by Bahadur et al. (2012). The apportionment was performed by a two step procedure, based on the assumption that $AAE^c$ followed a normal distribution featured by the parameters in Table 1. - Step 1. random extraction of $10^4$ $AAE^c$ for all species at all wavelengths. - Step 2. direct solution of the system of Angstrom equations. The steps 1 and 2 were repeated $10^4$ times for each retrieval in order to develop statistics of the $AAE^c$ combination that provides a solution to the system. The time series of median AAE c values was fairly stable over the measurement period, at both sites, except during an intense episode of dust transport, when the $AAE^c$ for BrC increased significantly and $AAE2^c$ for dust dropped (Figure S3). Both $AAE^c$ for BrC and $AAE2^c$ for dust values were on the tails of their respective distributions.*

The estimate of AAE for BC and BrC proceeds from a combination of the methods presented by Bahadur et al. (2012), Cazorla et al. (2013) and Shin et al. (2019) and, for both species, retrievals with dust were removed prior the computation of the AAE. However AAE for these species have some variability, described by their median and their median absolute deviation, and to solve the equation system we explored the whole distribution. The median and the median absolute deviation were preferred to mean and standard deviation, because they are more robust to outliers. A comment regarding BC and eBC was added to the RM, as follows (line 363): *It is worth noting that $AAE^c$ refers to BC and not to eBC since it proceeds from a direct estimate of the absorption wavelength dependence of aerosol particles while suspended*

*in air.*

22. *Line 295: What about Saturday's data? Also note a typo error for 'panel' here.*

Saturdays usually show a mixed signal between a holiday and a weekday (shops and schools are open on Sundays, most manufacturing activities will be reduced and many public offices are closed, but public services are available): since this feature is country-dependent it is stated more clearly in the RM (line 371) *Saturdays are excluded due to their mixed signal between a holiday and a weekday.*

23. *Lines 302-323: This part is basically a comparison with available literature. I think this extended text could be better summarized in a Table comparing results of this study with relevant literature (as done for the AERONET-based statistics).*

In the RM a new table (Table 4) was added to compare in-situ absorption from different literature studies and the corresponding original paragraph is now summarised in a few lines of text (lines 378 in the RM).

24. *The role of residential heating (other than BB) should be also commented when discussing the diurnal cycles of Figure 2. It is rarely mentioned within the text.*

An enhanced description of the PM emissions by BB was added in Section 2 Measurement site and methods (lines 153): "*More specific to non-industrial combustion (SNAP 2), most of buildings use compressed natural gas for both heating and cooking; consistently 99.4% of PM10 emissions by SNAP 2 are estimated to be produced by biomass combustion for domestic heating (ARPAE, 2020)."*

25. *If kept (see main comment III above), Section 3.2 title should be revised (e.g. Comparing absorption optical depth from remote sensing and in situ values).*

We agree with the suggestion and the title of the section was changed accordingly.

26. *Line 392: replace 'were' with 'was'*

Thanks, the RM was changed accordingly.

27. *Lines 400: Note that label 'a)' in the list is missing.*

Thanks, the RM was changed accordingly.

28. *Lines 400-405: To my opinion the results in Figure 6 clearly suggest one main point: for the species showing higher absorption at longer wavelengths (880, 675 nm) assumption of a*

*uniform absorption distribution through MLH is not valid. In fact, reasons for discrepancies listed by the authors as b) and c) would also affect results in the blue range (third panel), which is not the case.*

We agree with the reviewer that the different aerosol layering in the vertical is one of the main drivers of the results shown in Figure 6 of the original manuscript (OM), as clearly stated in lines 398-399 of the OM: *"These results suggests a very large accumulation of aerosols at the ground layer if compared to the atmospheric column and to the MLH, similar to previous observations in Milan during very stable atmospheric conditions"*. Why these layers are present is consistent with the points a) and b) listed in the manuscript: a) the main sources of LAA absorbing at longer wavelengths are ground sources (e.g. traffic, domestic heating) b) a ground thermal inversion is probably present in Modena, as shown by the closest radiosounding profile, i.e. the inversion was experimentally observed. The final result: ground emissions of LAA (mainly absorbing at longer wavelengths) are trapped within the inversion and do not mix vertically. The reviewer is correct in saying that the third panel should be affected as well by reason b), but since the dominant absorbing species in the blue range is not emitted at the ground (see also Figure 8 in the OM or 9 in the RM), the third panel represents mainly the absorption by long-range transported dust. Finally point c), regarding the uncertainty in the MLH estimate, holds true throughout the manuscript, along with the uncertainty in the AERONET inversion and the MA200 observations. Unfortunately MLH schemes of common meteorological models (e.g. IFS, WRF) fail in the description of very low thermal inversions (point c) such as those occurring in the Po valley. Nonetheless the Mean Error of 31 meters resulting from the comparison of ApAOD (at the UV range) and MLH is close to the uncertainty of ERA5 in Europe, which was shown to be ~51 m on average (~19 m as median). A discussion about the limitation of the MLH estimate by standard numerical weather prediction model (e.g. IFS, WRF) is now present in the RM (lines 228).

29. *Also note that definition and use of ApAOD is critical. First, it has the dimensions of a length; therefore the naming of this quantity seems not particularly suitable. Additionally, its definition assumes that AAOD=absinsitu x ApAOD, which might be not the case for example in cases of transport of elevated aerosol layers, as during dust transport events, in which aerosol layering might be different (e.g. AAOD=absinsitu x Hlayer1 + absaloft x HLayer2 )*

To avoid possible confusion with AOD, which is unitless, ApAOD was renamed Apparent Aerosol Optical Height (ApAOH), therefore more suitable with the dimensions of a length. We agree with the reviewer that the definition assumes a single homogeneous aerosol layer, which is the reason why this index was used to investigate the vertical mixing of LAA species, i.e. to test this hypothesis. As stated by the reviewer in this comment and previous ones, in case of multiple aerosol layers ApAOD assumption is not valid (see panels in Figure 6 OM or Figure 7 RM). On the other hand when the assumption is valid, ApAOD depth corresponds to MLH.

30. *Lines 425-434: It is certainly true that 'Differences in AAOD might originate from a different atmospheric layering and mixing' but this implies that statement at lines 425-433 needs to be rephrased. Although I understand this claim, in the current formulation the statement is not correct. In fact, the decrease in PM2.5 / PM10 ratio only indicates that some dust is reaching the ground, but is not telling anything about its vertical distribution.*

We agree with the reviewer that this sentence might sound ambiguous. Since no experimental vertical profiling of LAA is available in Modena, we built our hypothesis on ground-based columnar data and ground in-situ data. We agree that the $PM_{2.5}/PM_{10}$ ratio provides information

only about what is happening at ground height, nonetheless the results altogether (e.g. comment 29) point to a condition of well mixed distribution of dust across the MLH.

**References**

31. *References are often given in non-chronological order within the text. Please, check the journal instructions on this.*

    Multiple citations were ordered chronologically in the RM.

32. *Section 2.2 should include reference to O'Neill et al. papers, on which most AERONET almucantars retrievals are based on.*

    The following paper was now added to the RM (line 248): *O'Neill, N. T., Eck, T. F., Smirnov, A., Holben, B. N., and Thulasiraman, S.: Spectral discrimination of coarse and fine mode optical depth, Journal of Geophysical Research: Atmospheres, 108, https://doi.org/10.1029/2002JD002975, 2003*

**Figures**

33. *I think panels within Figures should be numbered (or labelled) and Figure captions should refer to this numbering (labelling) to be more readable. Please, check the journal instructions on this.*

    Figures are properly labelled where needed and follow journal guidelines (https://www.atmospheric-chemistry-and-physics.net/submission.html#figurestables). In Figure 2, having 16 panels, each panel is unambiguously identified by the axis label and the addition of a further label on the panel itself would simply decrease the readability of the plot. The same applies to Figure 3. For a better readability 1 cm of vertical space was added between the two panels of Figure S3 and the two panels of Figure S9 were split into two separate Figures (S8 and S9 in the RM).

34. *Figure S2: please improve color scale range, as in the current form it seems not optimal for the dataset presented (little variability among data dots color)*

    The aesthetics of the plot was improved, with a wider colour range, although with a single hue, in order to have color-blind friendly plot (see journal guidelines). Dots overlap and for that there is no easy solution, although some transparency was added in the revised version of this Figure.

**Tables**

35. *Table 1: why reporting medians and associated median absolute deviance? Generally, when reporting median values, associated percentiles (typically 25-75th) are rather included. Are you sure the same statistical parameters are reported in the referred literature? Given the wavelength dependence of the AAE, probably the table could be restructured separating values by spectral range, making the table more readable.*

    The referred literature generally reports mean +/- standard deviation, which is surely fine, but since these indexes can be severely affected by possible outliers, we preferred to use the

corresponding robust indexes for the center and the width of a distribution, i.e. the median and the median absolute deviation. The interquartile range would not compare smoothly to the standard deviation reported in the literature. In the RM, Table 2 has the rows sorted according to wavelengths.

36. *Table 2: specify wavelength rather than 'IR', for homogeneity with the text and Figures*
The RM was changed accordingly to this suggestion.

37. *Table 3: I would remove this one. As well documented in the text, most variables have a marked diurnal cycle, providing Median and interquartile values here could be misleading and hide important information on daily variability. I would rather suggest to insert Figure S4 in the main text, showing the diurnal cycles of the same quantities (as done for the atmospheric composition ones).*

The RM was changed accordingly to this suggestion: table 3 is now Table S1 in the Supplementary Material, and Figure S4 of the OM is now Figure 4 in the RM.

38. *Table A1 now is given before introducing Appendix A, and is not related to it*

LaTeX is an outstanding tool, but occasionally it is not straightforward to have tables/figures exactly where desired (unless maybe working directly on the class file): for this issue we hope for help from the typesetting team of the journal.

*TITLE: A further suggestion is to include reference to the study area in the manuscript title, as it seems a bit too general in its current form.*

In the RM the title was changed to "*Aerosol absorption by in-situ filter-based photometer and ground-based sun-photometer in a Po valley urban atmosphere*"

**References:**

ARPAE: Update on the Emilia Romagna regional inventory of atmospheric emissions for the year 2017 (in Italian), Tech. rep., ARPAE, 2020.

Evangeliou, N., Platt, S. M., Eckhardt, S., Lund Myhre, C., Laj, P., Alados-Arboledas, L., Backman, J., Brem, B. T., Fiebig, M., Flentje, H., Marinoni, A., Pandolfi, M., Yus-Dìez, J., Prats, N., Putaud, J. P., Sellegri, K., Sorribas, M., Eleftheriadis, K., Vratolis, S., Wiedensohler, A., and Stohl, A.: Changes in black carbon emissions over Europe due to COVID-19 lockdowns, Atmos. Chem. Phys., 21, 2675–2692, https://doi.org/10.5194/acp-21-2675-2021, 2021.